# Modelling Economic Growth, Carbon Emissions, and Fossil Fuel Consumption in China: Cointegration and Multivariate Causality

**DOI:** 10.3390/ijerph16214176

**Published:** 2019-10-29

**Authors:** Zhihui Lv, Amanda M. Y. Chu, Michael McAleer, Wing-Keung Wong

**Affiliations:** 1KLASMOE & School of Mathematics and Statistics, Northeast Normal University, Changchun 130024, China; luzh694@nenu.edu.cn; 2Department of Social Sciences, Education University of Hong Kong, Hong Kong 999077, China; amandachu@eduhk.hk; 3Department of Finance, Asia University, Taichung 41354, Taiwan; michael.mcaleer@gmail.com; 4Discipline of Business Analytics, University of Sydney Business School, Sydney, NSW 2006, Australia; 5Econometric Institute, Erasmus School of Economics, Erasmus University Rotterdam, 3062 Rotterdam, The Netherlands; 6Department of Economic Analysis and ICAE, Complutense University of Madrid, 28040 Madrid, Spain; 7Institute of Advanced Sciences, Yokohama National University, Yokohama, Kanagawa 240-8501, Japan; 8Department of Finance, Fintech Center, and Big Data Research Center, Asia University, Taichung 41354, Taiwan; 9Department of Medical Research, China Medical University Hospital, Taichung 40402, Taiwan; 10Department of Economics and Finance, Hang Seng University of Hong Kong, Hong Kong 999077, China

**Keywords:** energy consumption, economic growth, gross domestic product, CO_2_ emissions, granger causality, China

## Abstract

Most authors apply the Granger causality-VECM (vector error correction model), and Toda–Yamamoto procedures to investigate the relationships among fossil fuel consumption, ${\mathrm{CO}}_{2}$ emissions, and economic growth, though they ignore the group joint effects and nonlinear behaviour among the variables. In order to circumvent the limitations and bridge the gap in the literature, this paper combines cointegration and linear and nonlinear Granger causality in multivariate settings to investigate the long-run equilibrium, short-run impact, and dynamic causality relationships among economic growth, ${\mathrm{CO}}_{2}$ emissions, and fossil fuel consumption in China from 1965–2016. Using the combination of the newly developed econometric techniques, we obtain many novel empirical findings that are useful for policy makers. For example, cointegration and causality analysis imply that increasing ${\mathrm{CO}}_{2}$ emissions not only leads to immediate economic growth, but also future economic growth, both linearly and nonlinearly. In addition, the findings from cointegration and causality analysis in multivariate settings do not support the argument that reducing ${\mathrm{CO}}_{2}$ emissions and/or fossil fuel consumption does not lead to a slowdown in economic growth in China. The novel empirical findings are useful for policy makers in relation to fossil fuel consumption, ${\mathrm{CO}}_{2}$ emissions, and economic growth. Using the novel findings, governments can make better decisions regarding energy conservation and emission reductions policies without undermining the pace of economic growth in the long run.

## 1. Introduction

Fossil fuel consumption is an important topic because it is a symbol of modern civilization. Nonetheless, fossil fuel consumption could increase pollution and make a significant impact on the global natural ecosystem. On the other hand, overuse of fossil energy could make the problems of both energy shortage and climate change becomes more serious, threatening the sustainability of the Earth, and the development of humankind. Thus, to reduce fossil fuel consumption, control carbon dioxide emissions, and retain economic growth is a common task for countries worldwide.

In addition, academics have demonstrated the relationships between environmental pollutants and economic growth nexus. For example, Kuznets [1] has postulated that environmental degradation increases with per capita income at the beginning of economic growth, and decrease thereafter, which is known as the environmental Kuznets Curve (EKC). The emissions of ${\mathrm{CO}}_{2}$ have been used as a proxy for environmental pollution because ${\mathrm{CO}}_{2}$ emissions have been increasing sharply every year, thereby resulting in greenhouse gas effects and global warming, which affects the environment (see, for example, References [2,3,4,5]). Thus, it is interesting to study the relationships among fossil fuel consumption, environmental pollutants, and economic growth.

Many developed countries have been taking a lead on mitigating carbon emissions, providing financial resources, and transferring technology to developing countries to address ${\mathrm{CO}}_{2}$ emissions and climate change in recent decades. Tol [6,7] study the marginal cost and damage costs of ${\mathrm{CO}}_{2}$ emissions. To date, China has been cooperating with other countries and making an effort to control ${\mathrm{CO}}_{2}$ emissions and contribute to mitigating climate change.

In 2007, China introduced the National Climate Change Program, the first national policy to address climate change, and the first national program for developing countries in this field, to integrate climate change policies into the national development strategy. In 2009, the China State Council set the target of reducing 40–45% of its carbon intensity (unit GDP ${\mathrm{CO}}_{2}$ emissions) before 2020 [8]. In the “Thirteenth Five-Year Plan”, the Chinese Government introduced a range of targets and policies related to reducing both ${\mathrm{CO}}_{2}$ emissions and fossil fuel consumption, including reducing its carbon intensity by 18%, reducing energy intensity by 15%, increasing non-fossil energy accounts by 15%, and reducing a coal consumption cap target of 4.2 billion tons by 2020 (National Program on Climate Change, 2014–2020 [9].

Meanwhile, the Chinese Government confirmed that it will reach the peak of ${\mathrm{CO}}_{2}$ emissions by 2030 and undertake best efforts to reduce it. However, as a country with development via the path of “high energy consumption, high greenhouse gas emissions”, and once the highest total emissions in the world, China is now facing a huge challenge to reduce its fossil fuel consumption and ${\mathrm{CO}}_{2}$ emissions. In addition, as the per capita GDP in China is still quite low, the Chinese Government will continue to emphasize economic development as a top priority task for a long time. Thus, to realize the targets of energy conservation and emission reduction, while ensuring its economic development, it is important for the Chinese Government to examine the relationships among fossil fuel consumption, environmental pollutants, and economic growth, and look for new and alternative development paths.

There are many papers using different methodologies to study the relationships among fossil fuel consumption, ${\mathrm{CO}}_{2}$ emissions, and economic growth. To the best of our knowledge, the literature has applied the following methods, including the Toda and Yamamoto procedure, bivariate linear causality, multivariate linear causality, and vector error correction model (VECM) to study the relationships among fossil fuel consumption, ${\mathrm{CO}}_{2}$ emissions, and economic growth.

However, there are some limitations to the approaches that have been used in the literature. First, the tests may not be able to detect any nonlinear causal relationship among the variables. Second, the tests may not be able to measure the independent, dependent, and joint effects together, so that testing a series of single hypotheses is different from testing all hypotheses jointly. Even though some research in the literature has studied the joint effects among the variables and/or the error-correction terms by constructing the F-statistic, if the variables do not have any cointegration relationship, determining the joint effects can become problematic.

Nonethelss, there is evidence supporting the existence of nonlinear behaviour among fossil fuel consumption, ${\mathrm{CO}}_{2}$ emissions, and economic growth. For example, according to the changes in economic environment, changes in energy policies, and fluctuations in energy prices, Lee and Chang [10], show that economic events and regime changes can lead to structural changes in energy consumption patterns for a given time period, which creates a nonlinear rather than linear relationship between energy consumption and economic growth.

In order to circumvent these limitations, we use the Granger test proposed by Hiemstra and Jones [11], Bai et al. [12,13,14], and others to examine multivariate linear and nonlinear Granger causal relationships among fossil fuel consumption, ${\mathrm{CO}}_{2}$ emissions, and economic growth for China. This approach not only enables obtaining linear and nonlinear, but also examines the independent, dependent, and joint effects among fossil fuel consumption, ${\mathrm{CO}}_{2}$ emissions, and economic growth for China. These multivariate linear and nonlinear Granger causality findings are not only more interesting and thought-provoking than in the existing literature, but are also useful to government and independent policy makers in their decision making related to fossil fuel consumption, ${\mathrm{CO}}_{2}$ emissions, and economic growth.

In order to draw a better picture regarding the issue, together with using linear and nonlinear Granger causality analysis, we conduct the cointegration analysis by applying both the Johansen cointegration and ARDL bounds tests to examine the cointegration relationships among fossil fuel consumption, ${\mathrm{CO}}_{2}$ emissions, and economic growth for China. In addition, we strongly recommend that academics and practitioners use the multivariate nonlinear causality tests proposed by Bai et al. [12,13,14], as these methods can examine the multivariate nonlinear causality tests regardless of whether the cointegration relationship exists or not, while the other literature checks the multivariate causality depending on the existence of cointegration.

This paper provides many novel findings and inferences for China. For example, as shown in Table A1, only considering these three variables, we can conclude that if the government expands fossil fuel consumption, it will have two impacts on economic: first, it will expand GDP with immediate effect and, second, the increase in the rate of fossil fuel consumption will expand China’s economy to grow both linearly and nonlinearly in the future. The inference is useful for government and independent policy makers in their consideration of which policy they should choose to reduce fossil fuel consumption or carry out any policy regarding energy conservation so that its economy will be damaged as little as possible.

We also conclude that if the government carries out policies of energy conservation and emission reduction, it will significantly slow down economic growth with immediate effect, cause it to fall nonlinearly and not linearly in the future. We note that the current research is the first paper to draw such a significant conclusion. In addition, these conclusions show the advantages of combining cointegration and linear and nonlinear causality at the multivariate level. Thus, we recommend that academics, practitioners, and policy makers use both cointegration and causality analysis in multivariate settings in their analysis. The novel findings in the paper are useful for policy makers in relation to fossil fuel consumption, ${\mathrm{CO}}_{2}$ emissions, and economic growth. Using the novel findings, governments can reach better decisions regarding energy conservation and emissions reduction without undermining the pace of economic growth in the long run.

The remainder of the paper is organized as follows. Section 2 provides a review of the related extant literature. Section 3 presents the theoretical foundation. Section 4 describes the data and empirical methodology. Section 5 discusses the empirical results. Finally, Section 6 draws inference from the novel empirical findings, and proposes some policy implications. Section 7 concludes the paper.

## 2. Literature Review

In this section, we review research directions, methodologies, and findings in the literature on the relationships among energy consumption, ${\mathrm{CO}}_{2}$ emissions, and economic growth. The first strands in the literature focus on investigating the relationships between ${\mathrm{CO}}_{2}$ emissions and economic growth.

The relationship between ${\mathrm{CO}}_{2}$ emissions and economic growth is one of the most important research areas that have become the focus of numerous theoretical developments and many empirical applications. For example, Apergis and Payne [15], Soytas and Sari [16], Zhang and Cheng [17], and Menyah and Wolde-Rufael [18] examine the relationships between economic growth and ${\mathrm{CO}}_{2}$ emissions. They examine the relationships between environmental pollutants and economic growth nexus as the environmental Kuznets Curve (EKC). Selden and Song [19], Grossman and Krueger [20], List and Gallet [21], Stern and Common [22] and Song et al. [23] point out that the Kuznets Curve fits empirical cases well in many developed countries. However, Harbaugh et al. [24] and List et al. [25] show that the relationships between economic growth and environmental pollutants may be not robust for several emission pollutants.

In addition, different methods of analysis have been used to investigate the relationship between ${\mathrm{CO}}_{2}$ emissions and economic growth in different countries and regions. For example, Holtz-Eakin and Selden [26] and Heil and Selden [27] obtain a U-shaped EKC for ${\mathrm{CO}}_{2}$ per capita emissions by using parametric models with pooled data. Bertinelli and Strobl [28], Azomahou et al. [29], Bertinelli et al. [30], and Saboori et al. [31] investigate the relationships between ${\mathrm{CO}}_{2}$ emissions and economic growth by using nonparametric estimation techniques. Recently, Yeh and Liao [32] use an analytical tool of stochastic impacts on population, affluence, and technology to investigate the relationships between ${\mathrm{CO}}_{2}$ emissions and economic growth in Taiwan. Sadorsky [33], Heidari et al. [34], and Saidi and Hammami [35] apply the panel regression techniques to investigate the relationships between ${\mathrm{CO}}_{2}$ emissions and economic growth in other countries.

Many recent papers investigate the causality between ${\mathrm{CO}}_{2}$ emissions and economic growth, and obtain mixed results. For example, Salahuddin [36] finds no significant causality between ${\mathrm{CO}}_{2}$ emissions and economic growth in Gulf Cooperation Council countries. Alshehry and Belloumi [37] show bidirectional causality between ${\mathrm{CO}}_{2}$ emissions and economic growth in Saudi Arabia. Using the Granger-VECM approach, Ahmad et al. [38] show bidirectional causality between ${\mathrm{CO}}_{2}$ emissions and economic growth in the short run, and unidirectional causality from economic growth to ${\mathrm{CO}}_{2}$ emissions in the long run in Croatia. Cowan et al. [39], Wang et al. [40], Kasman and Duman [41], Bento and Moutinho [42], and Antonakakis et al. [43] include other variables in the analysis to investigate the causality between ${\mathrm{CO}}_{2}$ emissions and economic growth.

The relationship between energy consumption and economic growth is also one of the most important research areas in climatology, environmental science, and other areas after Kraft and Kraft [44] and others established the relationship between energy consumption and economic growth. For example, Tugcu et al. [45] show the renewable and non-renewable energy consumption and economic growth relationships for G7 countries. Bhattacharya et al. [46] investigate the effect of renewable energy consumption on economic growth for the top 38 countries. Empirical evidence shows that the relationships could be uni-directional, bi-directional causality, or no causality at all. For example, Stern [47,48], Masih and Masih [49], Soytas and Sari [50], Wolde-Rufael [51,52], Lee [53], Tsani [54], and Alam et al. [55] show that energy consumption causes economic growth. On the other hand, Mozumder and Marathe [56], Erdal et al. [57], and Payne [58] conclude that there exists bi-directional causal relationship between energy consumption and economic growth. Nonetheless, Altinay and Karagol [59], Jobert and Karanfil [60], Chiou-Wei et al. [61], Chontanawat et al. [62], and Halicioglu [63] conclude that there is no causality between energy consumption and economic growth for some countries.

In addition, global warming and the energy crunch have become very important topics in recent decades. This extends the relationship between energy consumption and ${\mathrm{CO}}_{2}$ emissions that has become a topical subject for academics and practitioners. Many studies have investigated the relationships and obtained mixed results. For example, Soytas et al. [64], Soytas and Sari [16], Lean and Smyth [65], and Alshehry and Belloumi [37] find the uni-directional causal relationships from ${\mathrm{CO}}_{2}$ emissions to energy consumption. Using simultaneous-equations models with panel data of 14 MENA (Middle East and North Africa) countries, Omri [66] documents uni-directional causality from energy consumption to ${\mathrm{CO}}_{2}$ emissions without any feedback effects. However, Halicioglu [63] finds bi-directional causality between ${\mathrm{CO}}_{2}$ emissions and energy consumption in Turkey, while Zhang and Cheng [17] find uni-directional causal relationships from energy consumption to ${\mathrm{CO}}_{2}$ emissions in China. Using the panel smooth transition regression (PSTR) model, Heidari et al. [34] show that energy consumption increases ${\mathrm{CO}}_{2}$ emissions when GDP capita exceeds 4686 USD or falls below 4686 USD in five ASEAN (Association of Southeast Asian Nations) countries. Other studies for the causal relationships include Ang [67], Apergis and Payne [15], Menyah and Wolde-Rutael [18], Alam et al. [55], and Pao and Tsai [68].

In the energy literature discussed above, cointegration and causality tests have been widely adopted to examine the underlying relationships among fossil fuel consumption, ${\mathrm{CO}}_{2}$ emissions, and economic growth. There are two principal cointegration tests: the Johansen cointegration test and the autoregressive distribution lag (ARDL) bounds test proposed by Pesaran et al. [69]. Fatai et al. [70], Narayan and Smyth [71], Wolde-Rufael [72], Narayan and Singh [73], Odhiambo [74], Acaravci and Ozturk [75] and Begum et al. [76] apply the ARDL bounds test, while Halicioglu [63], Odhiambo [77], and Chang [78] prefer to use Johansen’s maximum likelihood test in their analyses. Ang [3], Chandran et al. [79], and Lean and Smyth [65] use both approaches in their analyses. There are advantages and disadvantages for each method. For example, Gonzalo [80] uses the Monte Carlo approach to investigate the Johansen test, and concludes that the Johansen test performs better with the full information maximum likelihood procedure, and the test is appropriate when the identification of the exogenous variable is not possible a priori.

However, Odhiambo [77] finds that the Johansen test is sensitive to different sample sizes and different lag lengths. On the other hand, Narayan and Narayan [81,82] and Narayan and Smyth [83,84] demonstrate the advantages of using the ARDL test for small sample sizes and different lag lengths in that it does not demand all variables to be integrated of the same order. In addition, Harris and Sollis [85] point out that the approach always provides, not only unbiased estimates of the long-run model, but also valid t-statistics even when some of the regressors are endogenous. There are two widely-used causality tests, namely the Toda–Yamamoto procedure (TY procedure) and the error-correction modelling (ECM) procedure. The ECM procedure investigates causality from the short-run and long-run perspectives. Belloumi [86], Odhiambo [77], Chang [78], and Alam et al. [5] apply the ECM procedure, while Zhang and Chang [17], Tsani [54] and Rahman [87] use the TY procedure to examine the causality among economic growth, environmental pollutants, and energy consumption.

Similar to different cointegration tests, there are advantages and disadvantages between the TY method and ECM procedure. Toda [88] argues that the causality test employing Johansen-type ECM may suffer from severe pre-test biases as the pre-tests for cointegration ranks of this model are very sensitive to the values of the nuisance parameters in the finite sample. Toda and Yamamoto [89] propose a VAR (Vector autoregression) approach applied to any arbitrary level of integration. Zapata and Rambaldi [90] point out that the TY procedure has high power of the test in moderate to large samples. However, Yamada and Toda [91] show the FM-VAR test proposed by Phillips [92] and the ECM procedures are more powerful than the TY procedure.

However, in all related studies for China, as shown in Table A1 and Table A2 in the appendices, there are only a few papers that have investigated the relationships for China. In addition, there are some limitations on the approaches, including the TY and ECM procedures that examine the causality relationship in the literature. First, the tests may not be able to detect any nonlinear causal relationship among the variables. Second, the tests may not be able to measure the independent, dependent, and joint effects together. Testing a series of single hypotheses is different from testing all hypotheses jointly. Even though a few studies in the literature have considered the joint effects among the variables and/or the error-correction terms by constructing the F-statistic, if the variables do not have any cointegration relationship, determining the joint effects could become problematic.

Nonetheless, many studies, for example, Lee and Chang [10], support the existence of nonlinear behaviour among fossil fuel consumption, ${\mathrm{CO}}_{2}$ emissions, and economic growth, because economic events and regime changes can lead to structural changes in energy consumption patterns for a given time period. Consequently, this could create a nonlinear rather than linear relationship between energy consumption and economic growth. In order to circumvent these limitations, we use the Granger test proposed by Hiemstra and Jones [11], and Bai et al. [12,13,14] to examine multivariate linear and nonlinear Granger causal relationships among fossil fuel consumption, ${\mathrm{CO}}_{2}$ emissions, and economic growth for China. This approach not only enables linear and nonlinear relationships among the variables, but also examines the independent, dependent, and joint causalty effects among fossil fuel consumption, ${\mathrm{CO}}_{2}$ emissions, and economic growth for China.

In order to draw a more accurate analysis of the issue, together with linear and nonlinear Granger causality, we conduct cointegration analysis by applying both the Johansen cointegration and ARDL bounds tests to examine the cointegration relationships among fossil fuel consumption, ${\mathrm{CO}}_{2}$ emissions, and economic growth for China. The empirical findings are not only more interesting and thought-provoking than those obtained in the existing literature, but also more useful for government and independent private policy makers in their decision making related to fossil fuel consumption, ${\mathrm{CO}}_{2}$ emissions, and economic growth.

In addition, we strongly recommend academics and practitioners to use the multivariate nonlinear causality tests proposed by Bai et al. [12,13,14] because the methods allow scholars to examine the multivariate nonlinear causality tests regardless of the existence of cointegration relationships, while the other studies in the literature check the multivariate causality depending on the existence of cointegration relationships.

## 3. Theory

In this section, we present the theory underlying the relationships among economic growth, ${\mathrm{CO}}_{2}$ emissions and environmental pollutants. We first discuss the theory for the relationship between economic growth and ${\mathrm{CO}}_{2}$ emissions.

### 3.1. Economic Growth and $CO_{2}$ Emissions

Fossil fuel consumption is an important topic because it is a symbol of modern civilization. Nonetheless, fossil fuel consumption could increase pollution and make a significantly impact on global natural ecosystem. On the other hand, overuse of fossil energy could make the problems of both energy shortage and climate change become more serious, threatening the sustainability of the Earth and development of humankind. Thus, to reduce carbon dioxide emissions and control climate change, yet retain economic growth at present levels, is a common task for countries worldwide. Therefore, it is important to study the relationships between economic growth and carbon dioxide emissions.

Academics have demonstrated the relationships between environmental pollutants and the economic growth nexus. For example, Kuznets [1] has postulated that environmental degradation increases with per capita income at the beginning of economic growth, and decreases thereafter, known as the environmental Kuznets Curve (EKC). However, the relationships between economic growth and environmental pollutants may be not robust for the number of emission pollutants. The emissions of ${\mathrm{CO}}_{2}$ have been used as a proxy for environmental pollution because ${\mathrm{CO}}_{2}$ emissions have been increasing sharply every year, resulting in the greenhouse effect and global warming, and affecting economic growth. In this paper, we hypothesize Gross Domestic Product (GDP) to be a function of ${\mathrm{CO}}_{2}$ emissions, as shown in the following equation:(1)$$GDP_{t}=f_{1}\left(CO2_{t}\right)$$

### 3.2. Energy Consumption and Economic Growth

We turn to discuss the relationships between economic output and energy consumption. The relationship between energy consumption and economic growth is one of the most important areas in research in climatology, environmental science, and other areas after Kraft and Kraft [44] and others have established the relationship. However, the empirical evidence shows that their relationships could be uni-directional, bi-directional causality, or no causality at all.

The reasons that empirical results vary across different countries depend on the country’s development path, development stage, sources of energy used, energy policies applied, energy consumption levesl, institutional arrangementss and so on. The relationship between energy consumption and economic growth plays an important role for policy makers in both developing and developed countries. In addition, fossil fuel consumption, including coal consumption, oil consumption and natural gas consumption, is the main energy consumption at the present stage in China. We use $Total_{t}$ to denote fossil fuel consumption. Therefore, it is important and necessary to investigate the relationship between economic growth and energy consumption, as shown in the following:(2)$$Total_{t}=f_{2}\left(GDP_{t}\right)$$

### 3.3. Energy Consumption and $CO_{2}$ Emissions

Global warming and energy crunch are the main subjects in recentdecades because the causal relationship between energy consumption and ${\mathrm{CO}}_{2}$ emissions is a major problem. Some studies have found the uni-directional causal relationship from ${\mathrm{CO}}_{2}$ emissions to energy consumption, some document the uni-directional causal relationship from energy consumption to ${\mathrm{CO}}_{2}$ emissions, and others find bi-directional causality between ${\mathrm{CO}}_{2}$ emissions and energy consumption. In this paper, we hypothesize that the relationship between ${\mathrm{CO}}_{2}$ emissions and energy consumption satisfies the following equation:(3)$$CO2_{t}=f_{3}\left(Total_{t}\right).$$

Thus, from Equations (1)–(3), we explore the cointegration and causal relationships among economic output, energy consumption and ${\mathrm{CO}}_{2}$ emissions, especially cointegration and causality relationship, as expressed in the following:(4)$$GDP_{t}=f_{4}\left(CO2_{t},\;Total_{t}\right)$$
(5)$$Total_{t}=f_{5}\left(CO2_{t},\;GDP_{t}\right)$$
(6)$$CO2_{t}=f_{6}\left(GDP_{t},\;Total_{t}\right)$$

## 4. Data and Methodology

In this section, we present data and the methodology used in the empirical analysis.

### 4.1. Data

In the empirical analysis, we use annual time series data for China from the world development indicators data base [93], the World Bank [94], and Our World in Data [95]. The problem of the data is that countries with large populations, large economics, or both, tend to be the largest total fossil fuel consumption and $CO_{2}$ emissions countries. In order to circumvent this limit, per capita data are used. Using per capita Gross Domestic Product (GDP) to measure economic growth will be better in order to be able to respond to the situation of behavioral preferences and personal energy consumption.

All data covering the period 1965–2016 are used in the paper. We have a total of 53 annual data observations, and the dataset is selected in order to have the maximum number of observations depending on data availability in China. During this period, certain important economic and environment policies took place, so that it is meaningful to examine the period selected. Per capita GDP (current US$) is used as a proxy for economic growth and is denoted by $GDP_{t}$. Per capita data on fossil fuel consumption are measured as megawatt-hours, and denoted by $Total_{t}.$ Per capita ${\mathrm{CO}}_{2}$ emissions are denoted by $CO2_{t}$, and measured in metric tons to be used as a proxy for environmental pollution. All data covering the period 1965–2016 are used in natural logarithms to reduce heteroscedasticity, such that the elasticity can be interpreted, and the logarithmic variables have economic meaning as they are approximately the growth rates in the respective variables.

Another reason we use natural logarithms for all the variables is because the difference of the two consecutive natural logarithms of any variable is the return of the variable. In this paper, we will study the behavior of the returns for all the variables. We denote $GDP_{t}$, $Total_{t}$, and $CO2_{t}$ for $ln(GDP_{t})$, $ln(Total_{t})$ and $ln(CO2_{t})$ without causing confusion.

In order to illustrate the trend in each series in the same scale, we use 1965 as a base year. Figure 1 and Figure 2 exhibit all the series analyzed in the paper. From the figure, we find that all series grow quickly from late 1990s. Both fossil fuel consumption and ${\mathrm{CO}}_{2}$ emissions per capita drop slightly after 2013, and coal consumption decreases sharply after 2013. However, GDP, natural gas consumption, and oil consumption continue to increase after 2013. The change in fossil fuel consumption shows that the energy structural reform has made significant advances in China. However, from Figure 1 and Figure 2, it is observed that, in general, all the series are moving together with similar trends, so that there should be cointegration relationships among the variables.

Figure 3 presents the trends in the time series in logarithms. It is interesting to note from Figure 3 that the differences in the logarithmic series also seem to be moving together. Nevertheless, to draw such conclusion, one has to conduct a proper statistical analysis. We will discuss this in the next section.

### 4.2. Methodology

In this section, we will apply the Johansen cointegration test, autoregressive distributed lag (ARDL) bounds test, and linear and nonlinear causality test, and use the vector error correction model (VECM) or VAR model to examine the relationships among fossil fuel consumption ($Total_{t}$), $CO_{2}$ emissions ($CO2_{t}$), and GDP ($GDP_{t}$).

#### 4.2.1. Cointegration Test

An examination of the properties of the cointegration relationships among non-stationary time series is an important topic in economics. According to the cointegration method developed by Granger [96], Engle and Granger [97], Johansen [98,99], and Johansen and Juselius [100], if the series of fossil fuel consumption, ${\mathrm{CO}}_{2}$ emissions, and GDP are integrated of order d, denoted by I(d), each of the three series has a stationary, invertiable and ARMA representation after differencing d times. If there exists a cointegration relationship among the variables, then there could exist a cointegrating vector α (≠0) to satisfy $\alpha^{\prime }x_{t}\sim I\left(d-b\right),$
b>0 and $x_{t}={\left(CO2_{t},GDP_{t},\;Total_{t}\right)}^{\prime }$. If the variables satisfy these two conditions, we can apply a cointegration test to examine whether there exists any stable long-run relationship among fossil fuel consumption, ${\mathrm{CO}}_{2}$ emissions, and GDP.

##### Johansen Cointegration Test

There are two commonly used cointegration tests: the Johansen cointegration test and the autoregressive distribution lag (ARDL) bounds test. We first apply the Johansen cointegration test to check whether there exist any long-run relationships among fossil fuel consumption, ${\mathrm{CO}}_{2}$ emissions, and GDP. According to Johansen [98,99], Johansen and Juselius [100] and Johansen [101], we use the following unrestricted VAR model with p lags:(7)$$Z_{t}=\alpha +\overset{p}{\underset{i=1}{\sum }}\pi_{i}Z_{t-i}+\varphi D_{t}+\mu_{t}$$
where $Z_{t}$ is the vector of each component denoted by I(d), $Z_{t}={\left(CO2_{t},\;GDP_{t},Total_{t}\right)}^{\prime }$, $D_{t}$ denotes a vector of dummies, and $\pi_{i}\;\left(i=1,2,\ldots ,p\right)$ are matrices of lag polynomials. Model in Equation (8) can be presented in the following equaiton after differencing the time series of $Z_{t}$:(8)$$\Delta Z_{t}=\overset{p-1}{\underset{i=1}{\sum }}\Phi_{i}\Delta Z_{t-i}+\Pi Z_{t-1}+\varphi D_{t}+\varepsilon_{t}\;,$$
where Δ denotes the first difference operator. There exist *r* cointegration relationships if the rank of Π is greater than zero and less than 3, namely, R(Π)=r, (0<r<3), since, in this situation, Π=αβ′ and matrix α and β are the matrix of 3×r, and R(α)=R(β)=r. Thus, to examine the cointegration relation of $Z_{t}$ is equal to analysis the matrix of Π, this is the fundamental issue of the Johansen cointegration test.

There are two types of statistics of the Johansen cointegration test, namely the trace statistic and max-eigenvalue statistic. The null hypothesis of the trace test and the maximum eigenvalue test are the same, namely that the number of cointegration vectors is $r=r^{*}<k$, but the alternative hypotheses of two tests are not the same, one is r=k and the other is r=k+1. The two statistics are given, respectively, as follows:(9)$$\lambda_{max}\left(r,r+1\right)=-T\ln\left(1-\lambda_{r+1}\right)\;\;\;\mathrm{and}\;\lambda_{trace}\left(r\right)=-T\sum_{i=r+1}^{n}\ln(1-\lambda_{i})$$
where n is the maximum number of possible cointegrating vectors: n=3 in this paper, r=0, 1, 2,$\;\lambda_{1},\lambda_{2},\lambda_{3}$ are the eigenvalues and $\lambda_{1}>\lambda_{2}>\lambda_{3}$. For greater applicability forsmall samples, Cheung and Lai [102] argue that the critical values of the Johansen cointegration test should be scaled by T(T−np), where T denotes the effective number of observations, n the number of variables in the estimated system, and p is the lag parameter. Readers may refer to Johansen [98,99], Johansen and Juselius [100], Cheung and Lai [102] and Johansen [101] for further details of the Johansen cointegration test.

##### Autoregressive Distributed Lag (ARDL) Bound Test

In order to circumvent the limitation that the Johansen model could be sensitive to small sample sizes [77] and improve power of our results, we also use the ARDL test in the empirical analysis. The ARDL test proposed by Pesaran and Shin [103], and extended by Pesaran et al. [69], performs statistically better than the other cointegration tests, including the Johansen test and Engle and Granger [97] test. In addition, Narayan [104] provides sets of asymptotic critical values for the ARDL test for sample sizes from 30–80, such that the test is suitable for analysis with smaller sample sizes. Narayan and Narayan [81,82] and Narayan and Smyth [83,84] also demonstrate the advantages of using the ARDL test for small sample sizes.

Thus, we apply the ARDL test to investigate the long-run relationships for the following unrestricted error correction models (UECMs), in which Δ denotes the first difference operator:**Model 1**: $CO2_{t}\;and$$GDP_{t}$:(10)$$\Delta CO2_{t}=\;\alpha_{1}+\overset{n}{\underset{i=1}{\sum }}\beta_{i}\Delta CO2_{t-i}+\overset{n}{\underset{j=0}{\sum }}\beta_{j}\Delta GDP_{t-j}+\gamma_{1}CO2_{t-1}+\gamma_{2}GDP_{t-1}+\varepsilon_{1,t}$$(11)$$\Delta GDP_{t}=\;\alpha_{1}+\overset{n}{\underset{i=1}{\sum }}\beta_{i}\Delta GDP_{t-i}+\overset{n}{\underset{j=0}{\sum }}\beta_{j}\Delta CO2_{t-j}+\gamma_{1}GDP_{t-1}+\gamma_{2}CO2_{t-1}+\varepsilon_{2,t}$$**Model 2**: $CO2_{t}\;and\;Total_{t}$:(12)$$\Delta CO2_{t}=\;\alpha_{1}+\overset{n}{\underset{i=1}{\sum }}\beta_{i}\Delta CO2_{t-i}+\overset{n}{\underset{j=0}{\sum }}\beta_{j}\Delta Total_{t-j}+\gamma_{1}CO2_{t-1}+\gamma_{2}Total_{t-1}+\varepsilon_{1,t}$$(13)$$\Delta Total_{t}=\;\alpha_{1}+\overset{n}{\underset{i=1}{\sum }}\beta_{i}\Delta Total_{t-i}+\overset{n}{\underset{j=0}{\sum }}\beta_{j}\Delta CO2_{t-j}+\gamma_{1}Total_{t-1}+\gamma_{2}CO2_{t-1}+\varepsilon_{2,t}$$**Model 3:**$GDP_{t}\;and\;Total_{t}$:(14)$$\Delta GDP_{t}=\;\alpha_{1}+\overset{n}{\underset{i=1}{\sum }}\beta_{i}\Delta GDP_{t-i}+\overset{n}{\underset{j=0}{\sum }}\beta_{j}\Delta Total_{t-j}+\gamma_{1}GDP_{t-1}+\gamma_{2}Total_{t-1}+\varepsilon_{1,t}$$(15)$$\Delta Total_{t}=\;\alpha_{1}+\overset{n}{\underset{i=1}{\sum }}\beta_{i}\Delta Total_{t-i}+\overset{n}{\underset{j=0}{\sum }}\beta_{j}\Delta GDP_{t-j}+\gamma_{1}Total_{t-1}+\gamma_{2}GDP_{t-1}+\varepsilon_{2,t}.\;$$**Model 4:**$CO2_{t},\;GDP_{t}\;and\;Total_{t}$:(16)$$\Delta GDP_{t}=\;\alpha_{1}+\overset{n}{\underset{i=1}{\sum }}\beta_{i}\Delta GDP_{t-i}+\overset{n}{\underset{j=0}{\sum }}\beta_{j}\Delta Total_{t-j}+\overset{n}{\underset{k=0}{\sum }}\beta_{k}\Delta CO2_{t-k}+\gamma_{1}GDP_{t-1}+\gamma_{2}Total_{t-1}+\;\gamma_{3}CO2_{t-1}+\varepsilon_{1,t}$$(17)$$\Delta Total_{t}=\;\;\alpha_{1}+\overset{n}{\underset{i=1}{\sum }}\beta_{i}\Delta Total_{t-i}+\overset{n}{\underset{j=0}{\sum }}\beta_{j}\Delta GDP_{t-j}+\overset{n}{\underset{k=0}{\sum }}\beta_{k}\Delta CO2_{t-k}+\gamma_{1}Total_{t-1}+\;\gamma_{2}GDP_{t-1}+\gamma_{3}CO2_{t-1}+\varepsilon_{2,t}$$(18)$$\Delta CO2_{t}=\;\alpha_{1}+\overset{n}{\underset{i=1}{\sum }}\beta_{i}\Delta CO2_{t-i}+\overset{n}{\underset{j=0}{\sum }}\beta_{j}\Delta Total_{t-j}+\overset{n}{\underset{k=0}{\sum }}\beta_{j}\Delta GDP_{t-k}+\gamma_{1}CO2_{t-1}+\gamma_{2}Total_{t-1}+\gamma_{3}GDP_{t-1}+\varepsilon_{3,t}$$

The asymptotic distribution of the F-statistic under the null hypothesis $H_{0}:\gamma_{1}=\gamma_{2}\left(=\gamma_{3}\right)=0$ has a non-standard distribution, which depends on many factors, including the number of regressors, the sample sizes, among others. In addition, Pesaran et al. [69] and Narayan [104] provide two sets of asymptotic critical values for a given significance level for the fixed sample size. One set assumes that all regressors are I(0) and the other assumes that all regressors are I(1). They also provide two sets of bands generated by the critical values. If the calculated F-statistics fall outside the bounds, a conclusive decision can be obtained without knowing the order of integration of the regressors; if the calculated F-statistic is higher than the upper bounds of I(1), then the null hypothesis of no cointegration is rejected; and if the estimated F-statistic is smaller than the lower bounds of I(0), the null hypothesis of no cointegration cannot be rejected. The test becomes inclusive if the calculated F-statistic falls inside the bounds. Readers may refer to Pesaran and Shin [101], Pesaran et al. [67], Narayan [104], Narayan and Narayan [81,82], and Narayan and Smyth [83,84] for further information.

In this paper, since the time series of $Total_{t}$, $CO2_{t}$, and $GDP_{t}$ are non-stationary in levels and are in one order of cointegration, we use both the Johansen cointegration and ARDL tests to determine whether there is any cointegration relationship for any pair of variables, or for all variables together. The results shown in the next section confirm that there is a cointegration relationship for any pair of variables and for all the variables together.

Therefore, we suggest estimating the following cointegration equations:(19)$$CO2_{t}\;\;=\;\beta_{0}\;+\;\beta_{1}Total_{t}$$
(20)$$GDP_{t\;}\;=\;\alpha_{0}\;+\alpha_{1}Total_{t}$$
(21)$$GDP_{t\;}\;=\;\theta_{0}\;+\;\theta_{1}CO2_{t}$$
(22)$$CO2_{t}\;\;=\;\delta_{0}\;+\;\delta_{1}Total_{t}\;+\;\delta_{2}GDP_{t}$$

In addition, as proposed by Enders [105] and Feasel et al. [106], we will use these equations to estimate the long-run dynamic relationships and reconcile the short-run behaviour among fossil fuel consumption, ${\mathrm{CO}}_{2}$ emissions, and GDP.

#### 4.2.2. Granger Causality

In the literature of testing causality among the rate of fossil fuel consumption, the rate of ${\mathrm{CO}}_{2}$ emissions and economic growth, most authors have used the Granger causality test on the vector error correction model (VECM). This approach is called the Granger causality-VECM approach. Others have used the Toda-Yamamoto (TY) procedure, Hsiao’s Granger causality test, and Granger causality test. All these methods depend on the VAR model.

For example, the causality test between the rate of fossil fuel consumption and economic growth is equivalent to testing the following two hypotheses:$H_{0}^{1}$: the rate of fossil fuel consumption does not cause economic growth,$H_{0}^{2}$: economic growth does not cause the rate of fossil fuel consumption.

Granger [107] proposes Granger causality, while Granger [108] showed that the real world is “almost certainly nonlinear”. Baek and Brock [109] extend the linear causality test to the nonlinear causality test, which was further modified by Hiemstra and Jones [11] to process the nonlinearity issues. Compared with the linear and nonlinear causality test proposed by Hiemstra and Jones [11], the Granger causality-VECM, TY procedure, and the linear causality test all ignore any nonlinear behavior.

There is substantial evidence supporting the existence of nonlinear behaviour among the rate of fossil fuel consumption, the rate of ${\mathrm{CO}}_{2}$ emissions, and economic growth. For example, according to the changes in the economic environment, changes in energy policies, and fluctuations in energy prices, Lee and Chang [10] show that economic events and regime changes can lead to structural changes of energy consumption patterns for a given time period. This creates a nonlinear rather than linear relationship between energy consumption and economic growth.

Owing to a series of published policies for China, such as the National Climate Change Program in 2007; in 2009, the China State Council sets the target of reducing its carbon intensity (unit GDP ${\mathrm{CO}}_{2}$ emissions) by 40–45% before 2020, and “Thirteenth Five-Year Plan“. Thus, it is necessary to investigate both linear and nonlinear relationships among energy consumption, ${\mathrm{CO}}_{2}$ emissions, and economic growth.

In order to circumvent the limitations of both $H_{0}^{1}$ and $H_{0}^{2}$, we use the causality test between the rate of fossil fuel consumption and economic growth as an example to test the following hypotheses:$H_{0}^{1\prime }$: the rate of fossil fuel consumption does not cause economic growth if there are no linear and no nonlinear causalities from the rate of fossil fuel consumption to economic growth,$H_{0}^{2\prime }$: economic growth does not cause the rate of fossil fuel consumption if there are no linear and no nonlinear causalities from economic growth to the rate of fossil fuel consumption.

Chiou-Wei et al. [61] only provide evidence from linear and nonlinear bivariate Granger causality tests about energy consumption and economic growth, and do not include ${\mathrm{CO}}_{2}$ emissions. Thus, to the best of our knowledge, this paper is the first to apply (multivariate) linear causality to examine whether there is any uni-directional or bi-directional causality among energy consumption, ${\mathrm{CO}}_{2}$ emissions, and economic growth in China. The paper also applies the (multivariate) nonlinear causality test to examine whether there is any uni-directional or bi-directional causality among energy consumption, ${\mathrm{CO}}_{2}$ emissions, and economic growth in China, and examines whether there is any multivariate linear and nonlinear causality among energy consumption, ${\mathrm{CO}}_{2}$ emissions and economic growth.

According to the cointegration relationships among fossil fuel consumption,${\;\mathrm{CO}}_{2}$ emissions, and GDP, we use the linear and nonlinear Granger causality test to examine whether past information of the variable could contribute to the prediction of others in bivariate and multivariate situations. We will discuss the methodology of the linear and nonlinear causality in the following subsections.

##### Granger Linear Causality Test

In order to test the linear causality relationships among the difference series of $Total_{t}$, $CO2_{t}$, and $GDP_{t}$, there are commonly two models used–one is the Vector Autoregression (VAR) model and the other is the vector error correction model (VECM). Without loss of generality, we let two vector time series be $X_{t}=\left(X_{1,t},\ldots \;,\;X_{n_{1},t}\right)\prime$ and $Y_{t}=\left(Y_{1,t},\;\ldots ,Y_{n_{2},t}\right)\prime$. In testing the causality relationships between two vectors of I(1) time series, such as the variables of $Total_{t}$, $CO2_{t}$, and $GDP_{t}$ in this paper, we use the first differenced series, $\Delta X_{t}=\left(\Delta X_{1,t},\ldots ,\Delta X_{n_{1},t}\right)\prime$ and $\Delta Y_{t}=\left(\Delta Y_{1,t},\;\ldots ,\Delta Y_{n_{2},t}\right)\prime$, to construct the VAR model and VECM.

If two vectors of I(1) time series $X_{t}=\left(X_{1,t},\ldots \;,\;X_{n_{1},t}\right)\prime$ and $Y_{t}=\left(Y_{1,t},\;\ldots ,Y_{n_{2},t}\right)\prime$ are cointegrated, then we can construct the following VECM model to estimate the linear causality relationship:(23)$$\left(\begin{array}{c} \Delta X_{t}\\ \Delta Y_{t} \end{array}\right)=\left(\begin{array}{c} A_{x\left[n_{1}\times 1\right]}\\ A_{y\left[n_{2}\times 1\right]} \end{array}\right)+\left(\begin{array}{cc} A_{\mathrm{xx}}{\left(L\right)}_{\left[n_{1}\times n_{1}\right]} & A_{\mathrm{xy}}{\left(L\right)}_{\left[n_{1}\times n_{2}\right]}\\ A_{\mathrm{yx}}{\left(L\right)}_{\left[n_{2}\times n_{1}\right]} & A_{\mathrm{yy}}{\left(L\right)}_{\left[n_{2}\times n_{2}\right]} \end{array}\right)\left(\begin{array}{c} \Delta X_{t-1}\\ \Delta Y_{t-1} \end{array}\right)+\left(\begin{array}{c} \alpha_{x\left[n_{1}\times 1\right]}\\ \alpha_{y\left[n_{2}\times 1\right]} \end{array}\right)\cdot {\mathrm{ecm}}_{t-1}+\left(\begin{array}{c} e_{x,t}\\ e_{y,t} \end{array}\right)$$
to test whether there is any the linear causal relationship between the vectors of the stationary time series $\Delta X_{t}={\left(\Delta X_{1,t},\ldots ,\Delta X_{n_{1},t}\right)}^{\prime }$ and $\Delta Y_{t}={\left(\Delta Y_{1,t},\;\ldots ,\Delta Y_{n_{2},t}\right)}^{\prime }$, where $A_{x\left[n_{1}\times 1\right]}$ and $A_{y\left[n_{2}\times 1\right]}$ are two vectors of intercept terms, $A_{\mathrm{xx}}{\left(L\right)}_{\left[n_{1}\times n_{1}\right]}$, $A_{\mathrm{xy}}{\left(L\right)}_{\left[n_{1}\times n_{2}\right]}$, $A_{\mathrm{yx}}{\left(L\right)}_{\left[n_{2}\times n_{1}\right]}$ and $A_{\mathrm{yy}}{\left(L\right)}_{\left[n_{2}\times n_{2}\right]}$ are matrices of lag polynomials, ${\mathrm{ecm}}_{t-1}$ is lag one of the error correction term, and $\alpha_{x\left[n_{1}\times 1\right]}$ and $\alpha_{y\left[n_{2}\times 1\right]}$ are the coefficient vectors for the error correction term ${\mathrm{ecm}}_{t-1}$.

We examine whether there exists a causal relationship from $\Delta y_{t}\left(\Delta x_{t}\right)$ to $\Delta x_{t}\left(\Delta y_{t}\right)$, which is equivalent to examining whether to accept the null hypothesis $H_{01}$: $A_{\mathrm{xy}}\left(L\right)=0\left(H_{02}:{\;A}_{\mathrm{yx}}\left(L\right)=0\right)$ and/or $H_{03}:$
$\alpha_{x}=0\;\left(H_{04}:{\;\alpha }_{y}=0\right)$. If the null hypothesis is true, the statistic $\left(T-c\right)\left(\log\left|\sum_{0}\right|-\log\left|\sum \right|\right)$ follows an asymptotic $\chi^{2}$ distribution with the degree of freedom equal to the number of restrictions on the coefficients in the system. Readers can refer to Hiemstra and Jones [11], Bai et al. [12,13,14], and Chow et al. [110], and the references given therein, for further information regarding the test statistics.

##### Granger Nonlinear Causality Test

After applying the VECM to $Total_{t},CO2_{t}$, and $GDP_{t}$, we obtain their corresponding residuals $\left\{{\hat{\varepsilon }}_{1t}\right\}$ and $\left\{{\hat{\varepsilon }}_{2t}\right\}$ to test nonlinear causality based on the residual series. For simplicity, in this section we denote $X_{t}=\left(X_{1,t},\ldots ,X_{n_{1},t}\right)\prime$ and $Y_{t}=\left(Y_{1,t},\ldots ,Y_{n_{1},t}\right)\prime$ to be the corresponding residuals of any two vectors of variables to be examined. We let the lead vector and lag vector of a time series, say $X_{i,t},$ as follows: for $X_{i,t},\;i=1,\ldots ,n,$ the $m_{x_{i}}$-length lead vector, and the $L_{x_{i}}$-length lag vector of $X_{i,t}$ are:$$X_{i,t}^{m_{x_{i}}}\equiv \left(X_{i,t},X_{i,t+1},\ldots ,X_{i,t+m_{x_{i}}-1}\right),m_{x_{i}}=1,2,\ldots ,t=1,\;2,\;\ldots ,$$
$$X_{i,t-L_{x_{i}}}^{L_{x_{i}}}\equiv \left(X_{i,t-L_{x_{i}}},X_{i,t-L_{x_{i}}+1},\ldots ,X_{i,t-1}\right),L_{x_{i}}=1,\;2,\;\ldots ,\;t=L_{x_{i}}+1,L_{x_{i}}+2,\ldots ,$$
respectively. We denote $M_{x}=\left(m_{x1},\ldots ,m_{x_{n_{1}}}\right),$
$L_{x}=\left(L_{x1},\ldots ,L_{x_{n_{1}}}\right),{\;m}_{x}=\max\left(m_{x1},\ldots ,m_{n_{1}}\right),$ and $l_{x}=\max\left(L_{x1},\ldots ,L_{x_{n_{1}}}\right).$ The $m_{y_{i}}$-length lead vector, $Y_{i,t}^{m_{y_{i}}}$, the $L_{y_{i}}$-length lag vector, $Y_{i,t-L_{y_{i}}}^{L_{y_{i}}},$ of $Y_{i,t},$ and $M_{y},L_{y},{\;m}_{y},$ and $l_{y}$ can be defined similarly.

In order to test the null hypothesis, $H_{0}$, that $Y_{t}=\left(Y_{1,t},\ldots ,Y_{n_{1},t}\right)\prime$ does not strictly Granger cause $X_{t}=\left(X_{1,t},\ldots ,X_{n_{1},t}\right)\prime$ under the assumptions that the time series vector variables $X_{t}=\left(X_{1,t},\ldots ,X_{n_{1},t}\right)\prime$ and $Y_{t}=\left(Y_{1,t},\ldots ,Y_{n_{1},t}\right)\prime$ are strictly stationary, weakly dependent, and satisfy the mixing conditions stated in Denker and Keller [111], we first define the following four events, given that $m_{x},{\;m}_{y},{\;L}_{x},{\;L}_{y},$ and e>0:$$\left\{\|X_{t}^{M_{x}}-X_{s}^{M_{x}}\|<e\right\}\equiv \left\{\|X_{i,t}^{M_{x_{i}}}-X_{i,s}^{m_{x_{i}}}\|<e,\;\mathrm{for}\;\mathrm{any}\;i=1,\ldots ,n_{1}\right\};$$
$$\left\{\|X_{t-L_{x}}^{L_{x}}-X_{s-L_{x}}^{L_{x}}\|<e\right\}\equiv \left\{\|X_{i,t-L_{x_{i}}}^{L_{x_{i}}}-X_{i,s-L_{x_{i}}}^{L_{x_{i}}}\|<e,\;\mathrm{for}\;\mathrm{any}\;i=1,\ldots ,n_{1}\right\};$$
$$\left\{\|Y_{t}^{M_{y}}-Y_{s}^{M_{y}}\|<e\right\}\equiv \left\{\|Y_{i,t}^{m_{y_{i}}}-Y_{i,s}^{m_{y_{i}}}\|<e,\;\mathrm{for}\;\mathrm{any}\;i=1,\ldots ,n_{2}\right\};$$
and
$$\left\{\|Y_{t-L_{y}}^{L_{y}}-Y_{s-L_{y}}^{L_{y}}\|<e\right\}\equiv \left\{\|Y_{i,t-L_{y_{i}}}^{L_{y_{i}}}-Y_{i,s-L_{y_{i}}}^{L_{y_{i}}}\|<e,\;\mathrm{for}\;\mathrm{any}\;i=1,\ldots ,n_{2}\right\};$$
where ‖·‖ denotes the maximum norm which is defined as $\left\|X-Y\|=\max(\left|x_{1}-y_{1}\right|,\left|x_{2}-y_{2}\right|,\ldots ,\left|x_{n}-y_{n}\right|\right)$ for any two vectors $X=\left(x_{1},\ldots ,x_{n}\right)$ and $Y=\left(y_{1},\ldots ,y_{n}\right).$ The vector series $\left\{Y_{t}\right\}$ is said not to strictly Granger cause another vector series $\left\{X_{t}\right\}$ if:(24)$$\mathrm{Pr}\left(\|X_{t}^{M_{x}}-X_{s}^{M_{x}}\|<e|\|X_{t-L_{x}}^{L_{x}}-X_{s-L_{x}}^{L_{x}}\|<e,\|Y_{t-L_{y}}^{L_{y}}-Y_{s-L_{y}}^{L_{y}}\|<e,\right)\;=Pr\left(\|X_{t}^{M_{x}}-X_{s}^{M_{x}}\|<e|\|X_{t-L_{x}}^{L_{x}}-X_{s-L_{x}}^{L_{x}}\|<e\right),$$
where Pr(·|·) denotes conditional probability.

If the null hypothesis,$\;H_{0}$, is true, the test statistic:(25)$$\sqrt{n}\left(\frac{C_{1}\left(M_{x}+L_{x},L_{y},e,n\right)}{C_{2}\left(L_{x},L_{y},e,n\right)}\;\frac{C_{3}\left(M_{x}+L_{x},e,n\right)}{C_{4}\left(L_{x},e,n\right)}\right)$$
is distributed as $N\left(0,\sigma^{2}\left(M_{x},L_{x},L_{y},e\right)\right)$. When the test statistic is too far away from zero, we reject the null hypothesis. Readers may refer to Bai et al. [12,13,14] and Chow et al. [110] for the definitions of $C_{1}$, $C_{2}$, $C_{3}$, and $C_{4}$, and further information on the estimates of Equation (25).

### 4.3. Nonlinearity Test

In order to complement the nonlinear causality analysis, we conduct a nonlinearity test on the residuals from the VECM of $GDP_{t}$, $Total_{t}$, and $CO2_{t}$. It is necessary to conduct non-linearity analysis for the residuals because we believe that the residuals obtained from performing the linear causality test could cause nonlinear causality. In this paper, we conduct the nonlinearity test on $GDP_{t}$, $Total_{t}$, and $CO2_{t}$. In order to do so, we let $Y_{t}$ represents the residuals from each of the VECM of $GDP_{t}$, $Total_{t}$, and $CO2_{t}$.

The series $\left\{Y_{t}\right\}$ does not possess any nonlinearity if and only if, for any t, the law of corresponding residuals $\left\{Y_{t}\right\}$ satisfies $L\left(Y_{t}{|Y}_{t-1}\right)=L\left(Y_{t}\right),$ and we define $C_{1}\left(\tau \right)\equiv \mathrm{Pr}\left(Y_{t-1}<\tau ,Y_{t}<\tau \right)$, $C_{2}\left(\tau \right)\equiv \mathrm{Pr}\left(Y_{t-1}<\tau \right)$, and $C_{3}\left(\tau \right)\equiv \mathrm{Pr}\left(Y_{t}<\tau \right)$. As:(26)$$\mathrm{Pr}\left(Y_{t}<\tau {|Y}_{t-1}<\tau \right)=\frac{C_{1}\left(\tau \right)}{C_{2}\left(\tau \right)}$$

When testing the existence of the nonlinear of a sequence $\left\{Y_{t}\right\}$, we can test the following hypothesis:(27)$$H_{0}:\;\frac{C_{1}\left(\tau \right)}{C_{2}\left(\tau \right)}-C_{3}\left(\tau \right)=0$$

For a residual sequence $\left\{Y_{t}\right\}$, the dependence test statistic is given by:(28)$$T_{n}=\sqrt{n}\left(\frac{C_{1}\left(\tau ,n\right)}{C_{2}\left(\tau ,n\right)}-C_{3}\left(\tau ,n\right)\right)$$
where $C_{1}\left(\tau ,n\right)\equiv \frac{1}{n}\sum_{t=2}^{T}I_{\left(y_{t-1}<\tau \right)}\cdot I_{\left(y_{t}<\tau \right)}$, $C_{2}\left(\tau ,n\right)\equiv \frac{1}{n}\sum_{t=2}^{T}I_{\left(y_{t-1}<\tau \right)}$, $C_{1}\left(\tau ,n\right)\equiv \frac{1}{n}\sum_{t=2}^{T}I_{\left(y_{t}<\tau \right)}$, n=T−1, and *T* is the length of residual $\left\{Y_{t}\right\}$. Under this condition, if the residual $\left\{Y_{t}\right\}$ is *iid*, then the test statistic $T_{n}\to N\left(0,\sigma^{2}\left(\tau \right)\right)$, as n is large enough and the hypothesis:$H_{0}:\;\frac{C_{1}\left(\tau \right)}{C_{2}\left(\tau \right)}-C_{3}\left(\tau \right)=0$ is rejected at level α if |Tn|/σ^2(τ)>zα2. In this situation, series $GDP_{t}$, $Total_{t}$, and $CO2_{t}$ possess nonlinearity. The reader is referred to Hui et al. [112] and the references given therein for further details.

## 5. Empirical Analysis

In this paper, we apply the cointegration analysis, and multivariate linear and nonlinear causality tests, to examine whether there exists any long-term comovement, short-term impact, and multivariate linear and nonlinear causality among GDP, ${\mathrm{CO}}_{2}$ emissions, and fossil fuels consumption. Thereafter, we will check whether previous values of differences in the index can be used to predict future values of other variables. We start by examining the basic statistics to reveal the properties for all the variables examined in this paper. All the variables are defined in Section 3.

### 5.1. Basic Statistics

Before we conduct the analysis to examine cointegration and causality among the logarithms of $GDP_{t}$, $Total_{t}$, and $CO2_{t}$, we will discuss some basic descriptive statistics for each variable and display the results in Table 1.

Table 1 displays some basic descriptive statistics, including mean, variance, standard deviation (s.d.), medium, range, interquartile range (IQR), coefficient of variation (CV), skewness, excess kurtosis and Jarque–Bera (J–B) test of normality for the logarithms of $GDP_{t}$, $Total_{t}$, and $CO2_{t}$. From the table, we find that the means of all the variables are significantly positive at a 1% level, $CO2_{t}$ is more volatile than all the other variables, according to the value of CV, whereas $GDP_{t}$ is more dispersed than all the other variables, according to the value of range and IQR as $GDP_{t}$ has the highest range and the highest value of IQR among the variables.

It is also found that that skewness, excess kurtosis, and the Jarque–Bera (J–B) test of all the variables are not significantly different from zero, except $GDP_{t}$. The skewness of $GDP_{t}$ is significantly positive at the 10% level, implying that $GDP_{t}$ is skewed to the right. Moreover, the estimates of the skewness, kurtosis, and the Jarque–Bera (J–B) test show that all the time series are not rejected as being normally distributed, except $GDP_{t}$. In addition, by adapting both the univariate approach and outliers test, we conclude that there are no outlier or aberrant observations in the sample.

### 5.2. Stationarity Test Results

In order to test whether $GDP_{t}$, $Total_{t}$, and $CO2_{t}$ are cointegrated and have any causal relationship, we first conduct the unit root test to examine the integration order of the variables, and display the results in Table 2. The unit root and stationary tests, such as the Augmented Dickey–Fuller (ADF), Phillips–Perron (PP), DF-GLS, Kwiatkowski–Phillips–Schmidt–Shin (KPSS), Elliott, Rothenberg and Stock (ERS), Kapetanios–Shin (KS), and Kapetanios–Shin–Snell (KSS) tests, and the Leybourne–Newbold–Vougus (LNV) stationarity test, However, due to limitations in the sample size and the linear (UECMs, VECM, and VAR) models used in the paper, all the tests generally lead to similar conclusions.

We apply the ADF, PP, DF-GLS, KPSS, and ERS unit root tests in this paper to examine whether there are any unit roots, and examine the order of integration for all the variables used in this paper. The results are shown in Table 2. From the table, it is clear that all the tests draw the same conclusion, namely that all the series are nonstationary and their differences are stationary, inferring that all the variables are I(1).

### 5.3. Cointegration Test Results

As the results from Table 2 show that all the time series are I(1), we apply both the Johansen cointegration and ARDL tests to check whether there is any long-run relationship among $GDP_{t}$, $Total_{t}$, and $CO2_{t}$. To do so, we examine the following four relationships between: (a) GDP and fossil fuel consumption; (b) ${\mathrm{CO}}_{2}$ emissions and fossil fuel consumption; (c) ${\mathrm{CO}}_{2}$ emissions and GDP; and (d) fossil fuel consumption, ${\mathrm{CO}}_{2}$ emissions, and GDP. We conduct both the Johansen cointegration and ARDL tests for (a) to (d), and show the test results in Table 3 and Table 4, respectively.

From Table 3 and Table 4, we conclude the following: (a) there is one cointegration relationship between fossil fuel consumption and GDP, (b) there is one cointegration relationship between ${\mathrm{CO}}_{2}$ emissions and fossil fuel consumption, (c) there is one cointegration relationship between ${\mathrm{CO}}_{2}$ emissions and GDP, and (d) there is one cointegration relationships among GDP, ${\mathrm{CO}}_{2}$ emissions, and fossil fuel consumption. The existence of cointegration relationships implies that there is at least one linear combination among the variables that is stationary. We estimate all the cointegration relationships and show the results in Table 5.

From Table 5, we obtain the following estimated cointegration relationships:(29)$$CO2_{t}\;=\;-1.0857\;+\;0.9821Total_{t}$$
(−65.0275)  (109.8270)
(30)$$GDP_{t}\;=\;1.7370\;+\;2.3200Total_{t}$$
(9.3092)  (13.3722)
(31)$$GDP_{t}\;=\;4.3729\;+\;2.2786CO2_{t}$$
(45.3794)  (15.0931)
(32)$$CO2_{t}\;=\;1.2299\;+\;0.0573GDP_{t}+0.8717Total_{t}$$
(22.5919)  (2.7700)  (21.5338)

Equations (29)–(32) demonstrate the long-run relationship between ${\mathrm{CO}}_{2}$ emissions and fossil fuel consumption, between GDP and fossil fuel consumption, between ${\mathrm{CO}}_{2}$ emissions and GDP, and among ${\mathrm{CO}}_{2}$ emissions, fossil fuel consumption, and GDP, respectively. From Equation (29), we find that fossil fuel consumption has a significant positive impact on ${\mathrm{CO}}_{2}$ emissions, with a one percent increase in fossil fuel consumption leading to around a one percent increase in ${\mathrm{CO}}_{2}$ emissions. From Equation (30), we find that fossil fuel consumption has a significant positive impact on GDP, with a one percent increase in fossil fuel consumption significantly leading to around 2.32 percent increase in GDP.

From Equation (31), we find that ${\mathrm{CO}}_{2}$ emissions have a significantly positive impact on GDP, with a one percent increase in ${\mathrm{CO}}_{2}$ emissions significantly leading to around 2.2786 percent increase in GDP. In addition, all the equations show that GDP is more sensitive to fossil fuel consumption than ${\mathrm{CO}}_{2}$ emissions which are, in turn, more sensitive to fossil fuel consumption than GDP.

Similarly, from Equation (32), we find that both GDP and fossil fuel consumption have significant impacts on ${\mathrm{CO}}_{2}$ emissions. The estimates show that a one percent increases in fossil fuel consumption will significantly lead to around a 0.9 percent increase in ${\mathrm{CO}}_{2}$ emissions, and a one percent increase in GDP will significantly lead to around a 0.06 percent increase in ${\mathrm{CO}}_{2}$ emissions. However, the cointegration relationship alone cannot determine the existence of any causality relationship. In order to circumvent the limitation, we discuss the causality relationships among the variables in the next subsection.

### 5.4. Causality Test

The cointegration results show that there exist long-run relationships between ${\mathrm{CO}}_{2}$ emissions and fossil fuel consumption, between GDP and fossil fuel consumption, between ${\mathrm{CO}}_{2}$ emissions and GDP, and among ${\mathrm{CO}}_{2}$ emissions, fossil fuel consumption, and GDP, respectively. We now check whether there is any short-run linear and nonlinear causality among $\Delta GDP_{t}$, $\Delta CO2_{t}$, and $\Delta Total_{t}$ in both multivariate and bivariate situations. As the data used in the paper are expressed in logarithms, $\Delta CO2_{t}$, $\Delta Total_{t}$, and $\Delta GDP_{t}$ are the growth rates or returns in the corresponding variables.

We test whether there is any: (a) multivariate linear Granger causality, (b) bivariate linear Granger causality, (c) multivariate nonlinear Granger causality, and (d) bivariate nonlinear Granger causality among $\Delta GDP_{t}$, $\Delta CO2_{t}$, and $\Delta Total_{t}$. In addition, we examine whether there is any (e) nonlinearity associated with each variable. We explain (a) to (e) in the following subsections.

#### 5.4.1. Multivariate Linear Causality

We first conduct a multivariate linear Granger causality test among $\Delta CO2_{t}$, $\Delta Total_{t}$, and $\Delta GDP_{t}$ to check whether there is any linear causality among the variables, and whether any variable could influence another variable linearly. The results are presented in Table 6.

From Table 6, we conclude that there exists a strong multivariate linear causal relationship from both $\Delta GDP_{t}$ and $\Delta CO2_{t}$ to $\Delta Total_{t}$ at the 1% significance level. Similarly, there exists a strong multivariate linear causality relationship from both $\Delta GDP_{t}$ and $\Delta Total_{t}$ to $\Delta CO2_{t}$, while there is no significant multivariate linear causal relationship from both $\Delta CO2_{t}$ and $\Delta Total_{t}$ to $\Delta GDP_{t}$. The empirical findings imply that, at the current stage in the economic development of China: (a) the rate of fossil fuel consumption is significantly related with both economic growth and the rate of ${\mathrm{CO}}_{2}$ emissions; and (b) the rate of ${\mathrm{CO}}_{2}$ emissions is significantly related with both economic growth and the rate of fossil fuel consumption. However, (c) economic growth cannot be linearly predicted by the rates of fossil fuel consumption and ${\mathrm{CO}}_{2}$ emissions.

#### 5.4.2. Bivariate Linear Causality

The multivariate linear causality results cannot determine whether there is any linear causal relationship between the variables in each pair from three variables. Thus, we conduct the bivariate linear causality test between pairs of variables from $\Delta CO2_{t}$, $\Delta Total_{t}$, and $\Delta GDP_{t}$ in the next subsection, and present the results in Table 7.

From Table 7, we conclude that there exists a significant linear causal relationship between any pair of variables $\mathrm{from}\;\Delta CO2_{t}$, $\Delta Total_{t}$, and $\Delta GDP_{t}.$ All are significant at the 1% level, except the pairs $\Delta CO2_{t}$ to $\Delta GDP_{t}$ and $\Delta Total_{t}$ to $\Delta GDP_{t}$ that are significant at the 10% level. The empirical findings imply that the previous returns in any variable can be used to predict linearly another variable from among $\Delta CO2_{t}$, $\Delta Total_{t}$, and $\Delta GDP_{t}$.

However, the results from Table 6 and Table 7 show the existence of linear causality but not any nonlinear causal relationship among the three variables in multivariate and bivariate settings. It is well known that the existence of linear and nonlinear causal relationships are independent, as shown in, for example, Chiang et al. [113], Qiao et al. [114], and Chow et al. [110,115]. Thus, to examine more comprehensive relationships among the variables, we will examine nonlinear causality relationships in both multivariate and bivariate settings. In order to do so, we first examine whether there is any nonlinearity in the residuals from fitting linear causality, as discussed in the next subsection.

#### 5.4.3. Nonlinearity

Before we examine whether there is any nonlinear causal relationship among the variables in both multivariate and bivariate settings, we will examine whether there is any nonlinearity in the residuals from VECM and show the results in Table 8.

From Table 8, we conclude that there is no nonlinearity in the residuals from the VECM models, except the residuals of $\Delta CO2_{t}$ from $\Delta GDP_{t}$ at the 5% significance level. The results imply that the VECM models fit all the equations very well, except the VECM model of $\Delta CO2_{t}$ from $\Delta GDP_{t}.$ We discuss the issue in the next subsection.

#### 5.4.4. Multivariate Nonlinear Granger Causality

In order to test whether there is any nonlinear Granger causality among $\Delta CO2_{t}$, $\Delta Total_{t}$, and $\Delta GDP_{t}$, we check both multivariate and bivariate nonlinear causal relationships among $\Delta CO2_{t}$ and $\Delta Total_{t}$ to $\Delta GDP_{t}$. We first test whether there is any multivariate nonlinear causal relationship among $\Delta CO2_{t}$ and $\Delta Total_{t}$ to $\Delta GDP_{t}$, and show the results in Table 9.

The findings in Table 9 lead us conclude that there exist strong (5%) multivariate linear causality relationships: (a) from both $\Delta GDP_{t}$ and $\Delta CO2_{t}$ to $\Delta Total_{t}$, (b) from both $\Delta GDP_{t}$ and $\Delta Total_{t}$ to $\Delta CO2_{t}$, and (c) from both $\Delta CO2_{t}$ and $\Delta Total_{t}$ to $\Delta GDP_{t}$. The empirical findings imply that there exist at least one nonlinear component: (i) from economic growth and the rate of ${\mathrm{CO}}_{2}$ emissions that can strongly predict the rate of fossil fuel consumption, (ii) from economic growth and the rate of fossil fuel consumption that can strongly predict the rate of ${\mathrm{CO}}_{2}$ emissions, and (iii) from the rates of both ${\mathrm{CO}}_{2}$ emissions and fossil fuel consumption that can strongly predict economic growth.

#### 5.4.5. Bivariate Nonlinear Granger Causality

The results of the multivariate nonlinear causal relationships from both $\Delta CO2_{t}$ and $\Delta Total_{t}$ to $\Delta GDP_{t}$ discussed in Section 5.4.4 cannot lead us to conclude the existence of any nonlinear causal relationship between any pair of variables from $\Delta CO2_{t}$ and $\Delta Total_{t}$ to $\Delta GDP_{t}$. In order to draw conclusions on the existence of any bivariate nonlinear causal relationship among the variables, we will conduct a bivariate nonlinear causality test between any pair of $\Delta CO2_{t}$, $\Delta Total_{t}$, and $\Delta GDP_{t}$ in this subsection, and show the results in Table 10.

From Table 10, the empirical results show that there is not any significant nonlinear Granger causal relationship from $\Delta GDP_{t}$ to $\Delta Total_{t}$, from $\Delta GDP_{t}$ to $\Delta CO2_{t}$, and from $\Delta CO2_{t}$ to $\Delta Total_{t}$, while there are significant nonlinear Granger causal relationships from $\Delta Total_{t}$ to $\Delta GDP_{t}$ at the 10% level at Lag 4, from $\Delta CO2_{t}$ to $\Delta GDP_{t}$ at the 1% level at Lag 4, and from $\Delta Total_{t}$ to $\Delta CO2_{t}$ at the 10% level at both Lags 1 and 2. This implies that: (a) there is at least one nonlinear component in Lag 4 of the rate of fossil fuel consumption that can weakly predict the economic growth; (b) there is at least one nonlinear component in Lag 4 of the rate of ${\mathrm{CO}}_{2}$ emissions that can strongly predict economic growth; (c) there exists at least one nonlinear component in both Lags 1 and 2 of the rate of fossil fuel consumption that can weakly predict the rate of ${\mathrm{CO}}_{2}$ emissions; and (d) there is no other nonlinear component for any lag of any variable that can predict another variable from among $\Delta CO2_{t}$, $\Delta Total_{t}$, and $\Delta GDP_{t}.$

#### 5.4.6. Summary of Multivariate and Bivariate Cointegration and Causality

We can summarize the empirical findings in the paper by showing the multivariate (bivariate) cointegration and linear and nonlinear Granger causality results in Table 11 and Table 12.

We first discuss the summary of the multivariate cointegration and linear and nonlinear Granger causal relationships, as shown in Table 11. The table shows that there is a strong cointegration relationship among GDP, ${\mathrm{CO}}_{2}$ emissions, and fossil fuel consumption. There are also strong significant linear Granger causal relationships from both economic growth and the rate of ${\mathrm{CO}}_{2}$ emissions to the rate of fossil fuel consumption, and from both economic growth and the rate of fossil fuel consumption to the rate of ${\mathrm{CO}}_{2}$ emissions at the 1% significance level, respectively.

The empirical results also show that there exists a significant nonlinear Granger causal relationship from both economic growth and the rate of ${\mathrm{CO}}_{2}$ emissions to the rate of fossil fuel consumption at the 5% significance level, and a significant nonlinear Granger causal relationship from both economic growth and the rate of fossil fuel consumption to the rate of ${\mathrm{CO}}_{2}$ emissions at the 5% significance level. However, there exists significant nonlinear, but not significant linear, causality from the rates of both fossil fuel consumption and ${\mathrm{CO}}_{2}$ emissions to economic growth.

We now discuss the summary of the bivariate cointegration and linear and nonlinear Granger causal relationships, as shown in Table 12. We conclude that there are cointegration relationships between ${\mathrm{CO}}_{2}$ emissions and fossil fuel consumption, between GDP and fossil fuel consumption, and between ${\mathrm{CO}}_{2}$ emissions and GDP. It is also found that there exists a significant linear Granger causal relationship for any pair among the rate of ${\mathrm{CO}}_{2}$ emissions, the rate of fossil fuel consumption, and economic growth at the 1% significance level, except for the pairs from the rate of ${\mathrm{CO}}_{2}$ emissions to economic growth, and from the rate of fossil fuel consumption to economic growth, that are significant at the 10% level.

There is no nonlinear causal relationship of each pair of variables among the rate of fossil fuel consumption, the rate of ${\mathrm{CO}}_{2}$ emissions, and economic growth, except for the pairs from the rate of ${\mathrm{CO}}_{2}$ emissions to economic growth at the 1% significance level, and from the rate of fossil fuel consumption to both economic growth and ${\mathrm{CO}}_{2}$ emissions at the 10% significance level.

## 6. Inference

In this section, we draw inferences based on the empirical findings of the relationships among ${\mathrm{CO}}_{2}$ emissions, fossil fuel consumption, and economic growth in China. We first draw inferences from the cointegration relationships among ${\mathrm{CO}}_{2}$ emissions, fossil fuel consumption, and economic growth.

### 6.1. Cointegration Relationships among Fossil Fuel Consumption, $CO_{2}$ Emissions, and GDP

In this section, we first make the following conjectures:

**Conjecture** **1.**
*high fossil fuel consumption and high GDP move together positively.*


**Conjecture** **1′.**
*high $CO_{2}$ emissions and high GDP move together positively.*


In order to check whether Conjectures 1 and $1^{\prime }$ have any empirical validity, we establish the following hypotheses:$H_{0}^{3}:$ fossil fuel consumption and GDP are not positively cointegrated;$H_{1}^{3}:$ fossil fuel consumption and GDP are positively cointegrated;$H_{0}^{3\prime }:$${\mathrm{CO}}_{2}$ emissions and GDP are not positively cointegrated;$H_{1}^{3\prime }:$${\mathrm{CO}}_{2}$ emissions and GDP are positively cointegrated.

Using $GDP_{t}$ and $Total_{t}$ as proxies for both GDP and fossil fuel consumption, respectively, the results of both the Johansen cointegration and ARDL tests show that we do not reject both $H_{1}^{3}$ and $H_{1}^{3\prime }$, and conclude that there exist significant positive cointegration relationships between fossil fuel consumption and GDP, and between ${\mathrm{CO}}_{2}$ emissions and GDP. This leads to empirical support for both Conjectures 1 and $1^{\prime }$.

In addition, we examine whether it is possible to reduce both ${\mathrm{CO}}_{2}$ emissions and fossil fuel consumption, but not retard economic growth in China. In other words, under the condition of not sacrificing economic growth, is it plausible to establish public policy to yield energy conservation and emission reduction? In order to check whether this is possible, we establish the following conjecture:

**Conjecture** **2.**
*It is possible to reduce both $CO_{2}$ emissions and fossil fuel consumption, but not retard economic growth in China.*


In order to check whether Conjecture 2 has empirical support, we establish the following hypothesis:$H_{1}^{4}:$$CO_{2}$*emissions, fossil fuel consumption, and GDP* are not cointegrated.

The findings from Table 3 and Table 4 lead us conclude that $H_{1}^{4}$ is rejected, implying that there exists a significant cointegration relationship among fossil fuel consumption, ${\mathrm{CO}}_{2}$ emissions, and economic growth, which implies that Conjecture 2 is rejected.

Academics, practitioners, and policy makers are interested in knowing whether the policy of both energy conservation and emission reduction should be implemented. They are also interested in knowing whether regulating policies for energy conservation and emission reduction will affect economic growth. The empirical analysis suggests that implementing public policy of energy conservation and emission reduction will retard economic growth.

In the next subsection, we discuss inferences from the causal relationships among the rate of fossil fuel consumption, the rate of ${\mathrm{CO}}_{2}$ emissions, and economic growth, and discuss inferences from both the cointegration and causal relationships among fossil fuel consumption, ${\mathrm{CO}}_{2}$ emissions, and economic growth.

### 6.2. Causal Relationships among Rate of Fossil Fuel Consumption, Rate of $CO_{2}$ Emissions, and Economic Growth

We now draw inferences from the causal relationships among ${\mathrm{CO}}_{2}$ emissions, fossil fuel consumption, and economic growth in China.

#### 6.2.1. The Advantages of Using Both Linear and Nonlinear Causality

We first discuss the advantages of using both linear and nonlinear causality. In order to do so, we test the following hypotheses:$H_{0}^{5}$: the rate of fossil fuel consumption does not cause economic growth;$H_{0}^{6}$: the rate of ${\mathrm{CO}}_{2}$ emissions does not cause economic growth.

If one uses the Granger causality-VECM or TY procedure to test the above hypotheses, one could only obtain the information of a linear causal relationship between the rate of ${\mathrm{CO}}_{2}$ emissions and economic growth, and between the rate of fossil fuel consumption and economic growth. In this paper, we modify the above hypotheses, as follows:$H_{0}^{5^{\prime }}$: the rate of fossil fuel consumption does not cause economic growth if there is no linear or nonlinear causality from the rate of fossil fuel consumption to economic growth,$H_{1}^{5^{\prime }}$: the rate of fossil fuel consumption causes economic growth if there exist any linear and/or nonlinear causality from the rate of fossil fuel consumption to economic growth,$H_{0}^{6^{\prime }}$: the rate of ${\mathrm{CO}}_{2}$ emissions does not cause economic growth if there is no linear and no nonlinear causality from the rate of ${\mathrm{CO}}_{2}$ emissions to economic growth,$H_{1}^{6^{\prime }}$: the rate of ${\mathrm{CO}}_{2}$ emissions causes economic growth if there exist any linear and/or nonlinear causality from the rate of ${\mathrm{CO}}_{2}$ emissions to economic growth.

According to the results displayed in Table 7, Table 8, and Table 10 (and from Table 12), $H_{0}^{5^{\prime }}$ is rejected, $H_{1}^{5^{\prime }}$ is not rejected, and we conclude that the rate of fossil fuel consumption causes economic growth both linearly and nonlinearly at the 5% significance level. Thus, if the government expands the rate of fossil fuel consumption, it will significantly cause economic growth both linearly and nonlinearly in the future. Similarly, from Table 7, Table 8, and Table 10 (and from Table 12), one can reject $H_{0}^{6^{\prime }}$, and not reject $H_{1}^{6^{\prime }}$, and conclude that the rate of ${\mathrm{CO}}_{2}$ emissions causes economic growth linearly at the 10% level, and causes economic growth nonlinearly at the 1% level, implying that if the government expands the rate of ${\mathrm{CO}}_{2}$ emissions, it will significantly cause economic growth both linearly and nonlinearly in the future, with the nonlinear effect being stronger.

The empirical findings suggest public policymakers should examine any linear and nonlinear effects from the rates of both fossil fuel consumption and ${\mathrm{CO}}_{2}$ emissions that could cause economic growth, so that public policymakers could adopt better policies in both energy conservation and emission reduction, while expanding economic growth. This demonstrates the empirical advantages of using both linear and nonlinear causality tests.

The findings and the inferences drawn from them provide additional information compared with those who apply only one of linear or nonlinear causality test in their analyses. We note that many studies, such as Chiang et al. [113], Qiao et al. [114], and Chow et al. [110,115], find only linear causality between some pairs of variables, only nonlinear causality between other pairs of variables, both linear and nonlinear causalities between some other pairs of variables, and no linear or nonlinear causality between the remainder of the pairs. Thus, in this paper, we conjecture that the findings of linear and nonlinear causalities are independent and suggest application of both linear and nonlinear causality tests in all future empirical analyses.

#### 6.2.2. The Advantages of Using Multivariate Causality

Next we discuss the advantages of using multivariate linear and nonlinear causality tests. In order to do so, we compare the results of testing hypotheses $H_{0}^{5^{\prime }},\;H_{1}^{5^{\prime }},\;H_{0}^{6^{\prime }}\mathrm{and}\;H_{1}^{6^{\prime }}$ discussed in Section 6.2.1 with the results of testing the following hypotheses:$H_{0}^{7^{\prime }}:$ the rates of both fossil fuel consumption and ${\mathrm{CO}}_{2}$ emissions do not cause economic growth if there is no linear and no nonlinear causality from the rates of both fossil fuel consumption and ${\mathrm{CO}}_{2}$ emissions,$H_{1}^{7^{\prime }}$: the rates of both fossil fuel consumption and ${\mathrm{CO}}_{2}$ emissions cause economic growth if there is any linear and/or nonlinear causality from the rates of both fossil fuel consumption and ${\mathrm{CO}}_{2}$ emissions to economic growth.

The conclusions drawn from testing hypotheses $H_{0}^{5^{\prime }},\;H_{1}^{5^{\prime }},\;H_{0}^{6^{\prime }}\mathrm{and}\;H_{1}^{6^{\prime }}$ have been discussed in Section 6.2.1. Now we turn to discuss the conclusions from testing $H_{0}^{7^{\prime }}$ and $H_{1}^{7^{\prime }}.$ According to the results shown in Table 6 and Table 9 (and from Table 11), $H_{0}^{7^{\prime }}$ is rejected, $H_{1}^{7^{\prime }}$ is not rejected, and so we conclude that the rates of both fossil fuel consumption and ${\mathrm{CO}}_{2}$ emissions cause economic growth because there exists nonlinear but not linear causality from the rates of both fossil fuel consumption and ${\mathrm{CO}}_{2}$ emissions to economic growth in China. This suggests that, if the government reduces the rates of both fossil fuel consumption and ${\mathrm{CO}}_{2}$ emissions simultaneously, it will significantly cause economic growth in China nonlinearly. In other words, if the government institutes any public policy regarding both energy conservation and emission reduction, it will lead to China’s economy slowing down significantly.

The inferences drawn from testing $H_{0}^{7^{\prime }}$ and $H_{1}^{7^{\prime }}$ show that there is no linear and no nonlinear causality from the rates of both fossil fuel consumption and ${\mathrm{CO}}_{2}$ emissions to economic growth. This is different from the conclusion drawn from testing hypotheses $H_{0}^{5^{\prime }},\;H_{1}^{5^{\prime }},\;H_{0}^{6^{\prime }}\mathrm{and}\;H_{1}^{6^{\prime }}$ that, if the government expands the rate of fossil fuel consumption or expands the rate of ${\mathrm{CO}}_{2}$ emissions separately, it will significantly cause economic growth to expand both linearly and nonlinearly in the future. This demonstrates the advantages of using both multivariate linear and nonlinear causality tests that can draw conclusions that using both bivariate linear and nonlinear causality tests cannot draw.

### 6.3. Cointegration and Causality Relationship among Fossil Fuel Consumption, $CO_{2}$ Emissions, and Economic Growth

We now discuss the advantages of using both cointegration and causality analysis to examine the relationships among fossil fuel consumption, ${\mathrm{CO}}_{2}$ emissions, and economic growth.

#### 6.3.1. The Advantages of Using Both Cointegration and Causality

We first discuss the advantages of using both bivariate cointegration and causality analyses to examine the relationships among fossil fuel consumption, ${\mathrm{CO}}_{2}$ emissions, and economic growth. In order to do so, we test both $H_{0}^{3}$ and $H_{1}^{3}$, as stated in Section 6.1, and both $H_{0}^{5^{\prime }}$ and $H_{1}^{5^{\prime }},\;\mathrm{as}$ stated in Section 6.2.

According to the discussion in Section 6.1, we do not reject $H_{1}^{3}$ and conclude that there exists a significant positive cointegration relationship between fossil fuel consumption and GDP, implying that fossil fuel consumption and GDP are moving together positively and, if the government expands fossil fuel consumption, it will significantly cause GDP with immediate effect.

On the other hand, according to the discussion in Section 6.2, we do not reject $H_{1}^{5^{\prime }}$ and conclude that the rate of fossil fuel consumption causes economic growth, both linearly and nonlinearly, implying that if the government expands the rate of fossil fuel consumption, it will significantly lead to growth in China’s economy both linearly and nonlinearly in the future. Thus, if we apply both bivariate cointegration and causality analyses to examine the relationship between fossil fuel consumption and economic growth, we will conclude that fossil fuel consumption and GDP are moving together positively, and the rate of fossil fuel consumption will cause economic growth both linearly and nonlinearly. This further suggests that, if the government expands fossil fuel consumption, then fossil fuel consumption will have two impacts on economic growth: first, it will expand GDP with immediate effect and, second, increases in the rate of fossil fuel consumption will significantly lead China’s economy to grow both linearly and nonlinearly in the future.

On the other hand, if the government reduces fossil fuel consumption or carries out any public policy of energy conservation, it will significantly cause China’s economy to slow down with immediate effect, and its rate will significantly lead China’s economy to decline further, both linearly and nonlinearly, in the future. This inference is useful for government and public policy makers in their consideration of policies they should choose to reduce fossil fuel consumption, or carry out any policies regarding energy conservation, so that the economy will be negatively impacted as little as possible. Similar inferences can be drawn between ${\mathrm{CO}}_{2}$ emissions and economic growth.

#### 6.3.2. The Advantages of Using Both Multivariate Cointegration and Causality

We now discuss the advantages of using both multivariate cointegration and causality analysis to examine the relationships among fossil fuel consumption, ${\mathrm{CO}}_{2}$ emissions, and economic growth. In order to do so, we test $H_{0}^{4}$, as stated in Section 6.1, and $H_{0}^{7^{\prime }}$ and $H_{1}^{7^{\prime }}$, as stated in Section 6.2.

According to the discussion in Section 6.1, the empirical findings lead us to reject $H_{0}^{4}$ and conclude that there exists a significant cointegration relationship among fossil fuel consumption,$\;{\mathrm{CO}}_{2}$ emissions, and GDP. On the other hand, according to the discussion in Section 6.2.2, the findings lead us to conclude that the rates of both fossil fuel consumption and ${\mathrm{CO}}_{2}$ emissions causes economic growth because there is nonlinear, but not linear, causality from the rates of both fossil fuel consumption and ${\mathrm{CO}}_{2}$ emissions to economic growth in China.

Therefore, when applying both multivariate cointegration and causality analysis to examine the relationships among fossil fuel consumption, ${\mathrm{CO}}_{2}$ emissions, and economic growth, we conclude that if the government reduces both fossil fuel consumption and ${\mathrm{CO}}_{2}$ emissions simultaneously, there are two impacts on economic growth: first, it will affect economic growth with immediate effect, and second, the rates of both fossil fuel consumption and ${\mathrm{CO}}_{2}$ emissions will cause economic growth to fall nonlinearly rather than not linearly in the future. In other words, if the government carries out public policies regarding both energy conservation and emission reduction, it will significantly retard the economy with immediate effect, and cause economic growth to fall nonlinearly rather than linearly in the future.

## 7. Conclusions

Energy crunch and global warming have become very serious issues in recent decades. In order to circumvent the problem, some countries have developed new technology in order to reduce both ${\mathrm{CO}}_{2}$ emissions and energy consumption, while not restricting economic growth. Thus, controlling ${\mathrm{CO}}_{2}$ emissions, reducing fossil fuel consumption, and encouraging economic growth is an important task for all countries worldwide, including China.

In order to work in this direction, many studies in the literature of energy have used either a cointegration test or causality test to investigate the relationships among fossil fuel consumption, ${\mathrm{CO}}_{2}$ emissions, and economic growth in China. To the best of our knowledge, the literature has applied methods, such as the Toda and Yamamoto procedure, bivariate linear causality, multivariate linear causality, and VECM test to examine the causal relationships among fossil fuel consumption, ${\mathrm{CO}}_{2}$ emissions, and economic growth.

However, there are some limitations to these testing approaches. First, the tests may not be able to detect any multivariate nonlinear causal relationship among the variables. Second, the tests do not measure the independent, dependent and joint effects together, so that testing series of single hypothesis is different from testing all the hypotheses jointly. Even though some research in the literature has examined the joint effect and/or the error-correction term of the variables by constructing appropriate F-statistics, if the variables do not have cointegration relationships, the joint effects and long-term causality cannot be determined.

In order to circumvent the limitations of the approaches that have been used in the literature, this paper recommends applying multivariate nonlinear causality tests, together with cointegration and bivariate linear and nonlinear causality tests, to capture more inclusive information. The empirical findings are more interesting and thought-provoking than those in the extant literature.

In this paper, we have obtained many novel findings that are useful to government and public policy makers in their decision making related to fossil fuel consumption, ${\mathrm{CO}}_{2}$ emissions, and economic growth. For example, we find that there exists causality from the rate of ${\mathrm{CO}}_{2}$ emissions to economic growth for China. This finding is consistent with Halicioglu [63] for Turkey and Ghosh [116] for India.

A second new finding is that there exist not only linear joint causality from the rates of both fossil fuel consumption and ${\mathrm{CO}}_{2}$ emissions to economic growth and from both the rate of ${\mathrm{CO}}_{2}$ emissions and economic growth to the rate of fossil fuel consumption, but also nonlinear joint causality from both fossil fuel consumption and ${\mathrm{CO}}_{2}$ emissions to economic growth, and from both the rate of ${\mathrm{CO}}_{2}$ emissions and economic growth to the rate of fossil fuel consumption. These empirical findings lead to the conclusion that there exist joint causality from the rates of both fossil fuel consumption and ${\mathrm{CO}}_{2}$ emissions to economic growth, and from the rate of ${\mathrm{CO}}_{2}$ emissions and economic growth to the rate of fossil fuel consumption, more pervasive.

A third novel empirical finding is that there exists joint causality from the rates of both fossil fuel consumption and ${\mathrm{CO}}_{2}$ emissions to economic growth, and from both the rate of ${\mathrm{CO}}_{2}$ emissions and economic growth to the rate of fossil fuel consumption. However, there is no linear joint causality from both the rate of ${\mathrm{CO}}_{2}$ emissions and fossil fuel consumption to economic growth. The findings are consistent with those in Wang et al. [117].

A new fourth finding is that there exists nonlinear causality from the rate of ${\mathrm{CO}}_{2}$ emissions and fossil fuel consumption to economic growth, though there is no linear causality from the rate of ${\mathrm{CO}}_{2}$ emissions and fossil fuel consumption to economic growth. Chiou-Wei et al. [61] provide evidence from linear and nonlinear bivariate Granger causality testing about energy consumption and economic growth, but they did not include ${\mathrm{CO}}_{2}$ emissions in their analysis.

In addition, the cointegration analysis provides solid support in favour of the development path of “high fossil fuel consumption, high ${\mathrm{CO}}_{2}$ emissions, and high economic growth” over the past decade in China, by showing the long-run co-movement between fossil fuel consumption and GDP, and long-run term co-movement between fossil fuel consumption and ${\mathrm{CO}}_{2}$ emissions. The empirical findings in the paper provide public policymakers with a better understanding of the relationships among fossil fuel consumption, ${\mathrm{CO}}_{2}$ emissions, and economic growth, so that they could formulate improved energy and climate policies for China.

**Discussion** **1.**
*If the government regulates any policy for energy conversation, will it significantly cause China’s economy to slow down with immediate or future effect?*


According to the scope and analysis in the paper, applying both bivariate cointegration and causality analyses to examine the relationship between fossil fuel consumption and economic growth, we conclude that fossil fuel consumption and GDP move together positively, and the rate of fossil fuel consumption causes economic growth, both linearly and nonlinearly. This further implies that if the government expands fossil fuel consumption, it will have two impacts on economic growth: first, it will expand GDP with immediate effect, and second, an increase the rate of fossil fuel consumption will significantly lead China’s economy to grow both linearly and nonlinearly in the future.

On the other hand, if the government reduces fossil fuel consumption or carries out any policy of energy conversation, it will significantly cause China’s economy to slow down with immediate effect, and its rate will significantly lead China’s economy to fall further, both linearly and nonlinearly, in the future. The empirical findings also suggest that it is necessary to increase sustainable fossil fuel consumption to expand economic growth. The lack of smooth fossil fuel supply could become a serious constraint and undermine the pace of economic growth. This inference is useful for the government and public policy makers in their consideration of which policies they should choose to reduce fossil fuel consumption or regulate any policy of energy conversation so that economic growth will be retarded as little as possisble.

It might be argued that there are many other factors, such as rebound effect and energy, as inputs in order to power the economy. A strong trend in the reduction of energy consumption does not hamper economic development in China, with late-2015 to early-2019 providing a good empirical example. The findings only suggest that using our data and analysis support the conjecture and these conclusions. In recent years, China has experienced a transition period of economic development. It is obvious that China is developing a new economic path to replace the development path of “high fossil fuel consumption, high emission and high growth”. However, it could also be because, for example, our data set does not contain data with and without the development of new technology and advanced products, and/or our tools cannot analyze the effects of the development of new technology and advanced products.

**Discussion** **2.**
*If one conducts cointegration analysis and concludes that there is no significant cointegration relationship among fossil fuel consumption, $CO_{2}$ emissions, and economic growth, could this finding imply that it is possible to reduce both $CO_{2}$ emissions and fossil fuel consumption without leading to restricting economic growth in China?*


**Our answer is that it cannot do so**. It is essential to conduct causality analysis. If causality analysis concludes that reducing both fossil fuel consumption and ${\mathrm{CO}}_{2}$ emissions simultaneously will not cause economic growth, then we cannot conclude that the government could reduce both ${\mathrm{CO}}_{2}$ emissions and fossil fuel consumption without leading to retardation of economic growth in China. However, the results of the causality analysis reported in Table 10 suggest that reducing the rates of both fossil fuel consumption and ${\mathrm{CO}}_{2}$ emissions simultaneously will nonlinear but not linear cause economic growth.

Even if we were to apply cointegration analysis and conclude that there is a significant cointegration relationship among fossil fuel consumption,${\;\mathrm{CO}}_{2}$ emissions, and economic growth, as shown above, it will still be necessary to conduct causality analysis because the inferences drawn from the cointegration analysis are different from those drawn from the causality analysis, as discussed above.

The use of both cointegration and causality analysis in both bivariate and multivariate settings could not conclude that it is possible to reduce both ${\mathrm{CO}}_{2}$ emissions and fossil fuel consumption, while simultaneously not economic growth in China. However, there could be other analyses that might be used to draw the conclusion that it is possible to reduce both ${\mathrm{CO}}_{2}$ emissions and fossil fuel consumption without retarding economic growth. This is beyond the scope of the present paper, and is left foir future research.

**Discussion** **3.**
*Might it be possible to reduce both $CO_{2}$ emissions and fossil fuel consumption without retarding economic growth?*


By applying both multivariate cointegration and causality analysis to examine the relationships among fossil fuel consumption, ${\mathrm{CO}}_{2}$ emissions, and economic growth, we conclude that if the government reduces both fossil fuel consumption and ${\mathrm{CO}}_{2}$ emissions simultaneously, there are two impacts on economic growth: first, it will affect economic growth with immediate effect, and second, the rates of both fossil fuel consumption and ${\mathrm{CO}}_{2}$ emissions will cause economic growth to drop nonlinearly rather than linearly in the future. In other words, if the government regulates policies for both energy conservation and emissions reduction, it will significantly retard the economy with immediate effect and cause economic growth to fall nonlinearly and not linearly in the future.

The above discussion could draw several policy implications that are very important for public and private policy makers. For example, if the government reduces fossil fuel consumption or carries out any public policy relating to energy conservation, it will significantly cause China’s economy to slow down with immediate effect. In turn, the reduced rate of growth will significantly lead China’s economy to decline both linearly and nonlinearly in the future. This inference is useful for government and public policy makers in their decisions at to which policies they should choose: to reduce fossil fuel consumption, or carry out any policies to get energy conservation, so that the economy will have small negative repercussions as possible.

Therefore, when applying both multivariate cointegration and causality analysis to examine the relationships among fossil fuel consumption, ${\mathrm{CO}}_{2}$ emissions, and economic growth, we conclude that if the government reduces both fossil fuel consumption and ${\mathrm{CO}}_{2}$ emissions simultaneously, there are two impacts on economic growth: first, it will affect economic growth with immediate effect, and second, the rates of both fossil fuel consumption and ${\mathrm{CO}}_{2}$ emissions will cause economic growth to fall nonlinearly rather than linearly in the future. In other words, if the government carries out public policies regarding both energy conservation and emission reduction, it will significantly retard the economy with immediate effect, and cause economic growth to fall nonlinearly, though not linearly, in the future.

In addition, there have some arguments regarding the policies relating to both energy conservation and emission reduction. Some academics, practitioners, and public policy makers may suggest that consumers use energy more efficiently so that the government can reduce both ${\mathrm{CO}}_{2}$ emissions and fossil fuel consumption, without restricting economic growth in the long run. Some might also suggest that the Chinese Government should seek alternative clean energy sources, including solar, wind, hydro, wave, geothermal, bio-mass, bio-agricultural, aquacultural, and renewable energy that have fewer polluting effects, and do not harm the environment, while maintaining economic growth in the long run.

Others have suggested that the Chinese Government should develop new technologies and advanced products, and change from low technology production patterns to high technology production patterns. In so doing, China can reduce both ${\mathrm{CO}}_{2}$ emissions and fossil fuel consumption, and not restrict economic growth in the long run. However, the empirical findings in the paper do not support the conjecture that reducing both ${\mathrm{CO}}_{2}$ emissions and fossil fuel consumption does not lead to a reduction in economic growth. This does not imply that it is impossible to reduce both ${\mathrm{CO}}_{2}$ emissions and fossil fuel consumption and yet restrict any slowdown in economic growth.

On the contrary, the empirical findings suggest that, based on the data used in this paper, the empirical analysis does not support the conjecture. However, it could be because the data set either does not contain appropriate data that include the development of new technology and advanced products, or the present techniques that were used in this paper do not include the effects of the development of new technology and advanced products in the analysis, because such information is not contained in the dataset used for the empirical analysis.

In the paper, we investigated long-run equilibrium, short-run impacts, and causality relationships among fossil fuel consumption, ${\mathrm{CO}}_{2}$ emissions, and economic growth, by applying the cointegration test, and linear and nonlinear causality tests in the bivariate and multivariate settings.

Extensions of these empirical results would include other related variables. We note that sensitivity and uncertainty analyses examine how the uncertainty in the output of a mathematical model or system can be decomposed and allocated to different sources of uncertainty in its inputs [118]. As we only have one input of data, we did not conduct sensitivity and uncertainty analyses in this paper. Further research might be able to access different sources of data so that one could include sensitivity and uncertainty analyses in the empirical analysis.

Further research could also include other tools, for example, portfolio optimization (see, for example, References [119,120,121]), stochastic dominance (see, for example, References [122,123,124,125]), and risk measures (see, for example, References [126,127,128,129,130,131]) to analyze the relationships among fossil fuel consumption, ${\mathrm{CO}}_{2}$ emissions, and economic growth for China as well as other countries.

## Figures and Tables

**Figure 1 ijerph-16-04176-f001:**
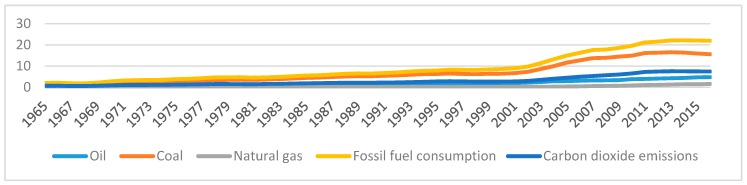
Time series plots of the variables used in this paper.

**Figure 2 ijerph-16-04176-f002:**
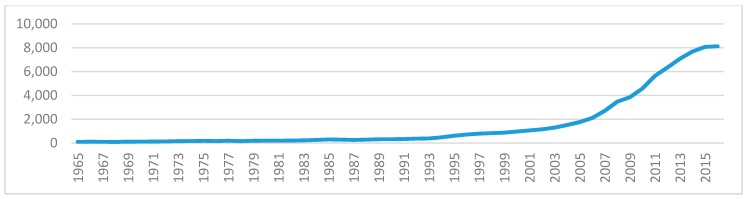
Time series plot of GDP per capita (current US$).

**Figure 3 ijerph-16-04176-f003:**
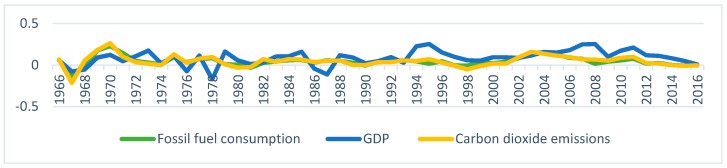
Time series plot for the difference in each variable (in logarithms).

**Table 1 ijerph-16-04176-t001:** Descriptive statistics for the variables.

Variables	Mean	Variance	S.d.	Medium	Range	IQR	CV	Skewness	Kurtosis	J–B
$GDP_{t}$	6.2959 ***	1.9486	1.3959	5.785129	4.4864	1.9937	0.2217	0.6010 *	−0.9530	4.9657 *
$CO2_{t}$	0.8249 ***	0.5267	0.7257	0.784208	2.5773	0.9277	0.8798	0.0653	−0.8655	1.3606
$Total_{t}$	1.9449 ***	0.5211	0.7219	1.892771	2.4836	0.9642	0.3712	0.0596	−0.9421	1.6374

This table reports the summary statistics including mean, variance, standard deviation (s.d.), medium, range, interquartile range (IQR), coefficient of variation (CV), skewness, excess kurtosis and Jarque–Bera (J–B) test for normality. *, ** and *** denote significance at the 10%, 5%, and 1% levels, respectively.

**Table 2 ijerph-16-04176-t002:** Unit root tests.

Variables	ADF Test		PP Test		DF-GLS		KPSS		ERS Test	
	Level	1st Difference	Level	1st Difference	Level	1st Difference	Level	1st Difference	Level	1st Difference
$CO2_{t}$	2.9019	−4.6426 ***	−2.2674	−4.2563 ***	−1.7346	−3.5359 ***	1.4160 ***	0.0551	0.8268	8.8529 ***
$GDP_{t}$	−1.2189	−5.2370 ***	−1.2747	−5.2555 ***	−1.085349	−3.437260 ***	0.2400 ***	0.5621	1.186	15.8102 ***
$Total_{t}$	−3.1225	−3.8970 ***	−2.0409	−3.2971**	−3.1822 *	−3.9227 ***	0.4772 ***	0.1337	0.7813	3.1511 ***

The table presents the results of Augmented Dickey–Fuller (ADF), Phillips–Perron (PP), DF-GLS, Kwiatkowski–Phillips–Schmidt–Shin (KPSS), Elliott, Rothenberg and Stock (ERS), Kapetanios–Shin (KS), and Kapetanios–Shin–Snell (KSS) tests, and the Leybourne–Newbold–Vougus (LNV) stationarity test. *, ** and *** denote significance at the 10%, 5%, and 1% levels, respectively.

**Table 3 ijerph-16-04176-t003:** Johansen Cointegration Test for $GDP_{t}$, $Total_{t}$, and $CO2_{t}$.

Variables	Hypothesized No. of Coinegrating Equations	Trace Statistic	Max-Eigen Statistic
${\mathrm{GDP}}_{t}\;and\;{\mathrm{Total}}_{t}$	None	16.8749 **	16.0207 **
	At most 1	0.8542	0.8542
$CO2_{t}\;and\;{\mathrm{Total}}_{t}$	None	30.9314 ***	28.4324 ***
	At most 1	2.4991	2.4991
$CO2_{t}\;and\;{\mathrm{GDP}}_{t}$	None	21.5744 ***	20.3308 ***
	At most 1	1.2435	1.2435
$CO2_{t},{\mathrm{GDP}}_{t}\;and{\;\mathrm{Total}}_{t}$	None	46.7777 ***	35.0982 ***
	At most 1	11.6795	10.1707

Note: *, ** and *** denote significance at 10%, 5%, and 1% levels, respectively.

**Table 4 ijerph-16-04176-t004:** ARDL bound test results for $GDP_{t}$, $Total_{t}$, and $CO2_{t}$

Dependent Variable				F-Statistics		Dependent Variable				F-Statistics		
$GDP_{t}\;and\;Total_{t}$						$CO2_{t}\;and\;Total_{t}$						
$GDP_{t}$				7.1840 ***		$CO2_{t}$				7.3135 ***		
$Total_{t}$				2.0756		$Total_{t}$				1.9180		
$\mathrm{CO}2_{t}\;and\;GDP_{t}$						$CO2_{t},GDP_{t}\;and\;\;Total_{t}$						
$CO2_{t}$				2338		$CO2_{t}$				4.9157 **		
$GDP_{t}$				6.5126 ***		$GDP_{t}$				2.0197		
						$Total_{t}$				1.7055		
Bound critical values												
	1% significance level				5% significance level				10% significance level			
	k=1		k=2		k=1		k=2		k=1		k=2	
n	I(0)	I(1)	I(0)	I(1)	I(0)	I(1)	I(0)	I(1)	I(0)	I(1)	I(0)	I(1)
50	5.503	6.240	4.695	5.758	3.860	4.440	3.368	4.178	3.177	3.653	2.788	3.513
55	5.377	6.047	4.610	5.563	3.790	4.393	3.303	4.100	3.143	3.670	2.748	3.495

Note: Asymptotic critical value bounds are obtained from Narayan [99]. k denotes the number of exogenous variables. *, ** and *** denote significance at 10%, 5%, and 1% levels, respectively.

**Table 5 ijerph-16-04176-t005:** Cointegration relationship for $CO2_{t}$ and $GDP_{t}$.

Cointegrating Equation:	$CO2_{t}$	$GDP_{t}$	$GDP_{t}$	$CO2_{t}$
$Total_{t}$	0.9821 ***	2.3200 ***		0.8717 ***
	(109.8270)	(13.3722)		(21.5338)
$CO2_{t}$			2.2786 ***	
			(15.0931)	
$GDP_{t}$				0.0573 ***
				(2.7700)
C	−1.0857 ***	1.7370 ***	4.3729 ***	1.2299 ***
	(−65.0275)	(9.3092)	(45.3794)	(22.5919)
Adj.R-squared	0.9967	0.8883	0.8921	0.9959
F-statistic	7452.8760 ***	194.7932 ***	202.5335 ***	5710.684 ***
ADF test for residual	−3.0546 ***	−4.1793 ***	−2.6642 ***	−2.3912 **

Note: *, ** and *** denote significance at the 10%, 5%, and 1% levels, respectively. The upper entries are the estimated coefficients, and the lower entries are T-statistics in ( ).

**Table 6 ijerph-16-04176-t006:** Multivariate Linear Granger Causality Test.

	$\Delta GDP_{t},\Delta CO2_{t}\to \Delta Total_{t}$	$\Delta Total_{t},\Delta CO2_{t}\to \Delta GDP_{t}$	$\Delta GDP_{t},\Delta Total_{t}\to \Delta CO2_{t}$
Lags	2	2	2
F-Stat	7.1280 ***	0.6737	9.5789 ***

*, ** and *** denote significance at 10%, 5%, and 1% levels, respectively, and the symbol Δ denotes first-order difference. The notation “→” indicates the direction of causality, such that “A → B” indicates causality from A to B.

**Table 7 ijerph-16-04176-t007:** Bivariate Linear Granger Causality Test.

	$\Delta GDP_{t}\to \Delta Total_{t}$	$\Delta Total_{t}\to \Delta GDP_{t}$	$\Delta GDP_{t}\to \Delta CO2_{t}$	$\Delta CO2_{t}\to \Delta GDP_{t}$	$\Delta Total_{t}\to \Delta CO2_{t}$	$\Delta CO2_{t}\to \Delta Total_{t}$
Lags	2	2	2	2	2	2
F-Stat	5.7921 ***	3.0693 *	8.5011 ***	2.5480 *	16.9340 ***	9.6853 ***

*, ** and *** denote significance at 10%, 5%, and 1% levels, respectively, and the symbol Δ denotes first-order difference. The notation “→” indicates the direction of causality, such that “A → B” indicates causality from A to B.

**Table 8 ijerph-16-04176-t008:** Nonlinearity Test.

Parameter	$\Delta CO2_{t}\;\mathrm{and}\;\Delta GDP_{t}$		$\Delta Total_{t}\;\mathrm{and}\;\Delta GDP_{t}$		$\Delta Total_{t}\;\mathrm{and}\;\Delta CO2_{t}$		$\Delta GDP_{t},\;\Delta Total_{t},\;\mathrm{and}\;\Delta CO2_{t}$		
	$\Delta CO2_{t}$	$\Delta GDP_{t}$	$\Delta Total_{t}$	$\Delta GDP_{t}$	$\Delta Total_{t}$	$\Delta CO2_{t}$	$\Delta CO2_{t}$	$\Delta GDP_{t}$	$\Delta Total_{t}$
e = 1	2.4057 **	0.4066	−0.4678	0.3109	0.1732	0.8133	−0.03420	0.5552	−0.0342
e = 1.5	1.4058	0.3873	0.3888	−0.2629	0.1261	1.1880	0.4965	0.9408	0.4967

*, ** and *** denote significance at 10%, 5%, and 1% levels, respectively. The symbol Δ denotes first-order difference.

**Table 9 ijerph-16-04176-t009:** Multivariate Nonlinear Granger Causality Test.

Lags	$\Delta GDP_{t},\Delta CO2_{t}\;\to \Delta Total_{t}$	$\Delta Total_{t},\Delta CO2_{t}\;\to \Delta GDP_{t}$	$\Delta GDP_{t},\Delta Total_{t}\to \Delta CO2_{t}$
1	−1.4705 *	1.0328	1.4537 *
2	−2.2147 **	1.9151 **	1.6529 **
3	−1.1186	0.1516	1.9197 **
4	−0.4571	−1.0864	0.5101

*, ** and *** denote significance at 10%, 5%, and 1% levels, respectively, and the symbol Δ denotes first-order difference. The notation “→” indicates the direction of causality, such that “A → B” indicates causality from A to B.

**Table 10 ijerph-16-04176-t010:** Bivariate Nonlinear Granger Causality Test.

Lags	$\Delta GDP_{t}\to \Delta Total_{t}$	$\Delta Total_{t}\to \Delta GDP_{t}$	$\Delta GDP_{t}\to \Delta CO2_{t}$	$\Delta CO2_{t}\to \Delta GDP_{t}$	$\Delta Total_{t}\to \Delta CO2_{t}$	$\Delta CO2_{t}\;\to \Delta Total_{t}$
1	−0.3083	−0.0994	0.6379	0.0383	1.2988 *	−0.4053
2	−0.4359	−0.5058	0.8255	−0.5622	1.3347 *	−0.4525
3	−0.5472	−0.3906	0.3553	−0.3940	0.6415	0.2063
4	−1.0020	−1.3324 *	0.0986	3.7040 ***	0.5571	0.0123

*, ** and *** denote significance at 10%, 5%, and 1% levels, respectively, and the symbol Δ denotes first-order difference. The notation “→” indicates the direction of causality, such that “A → B” indicates causality from A to B.

**Table 11 ijerph-16-04176-t011:** Summary of multivariate cointegration and causality results.

Independent Variable	Dependent Variable	Cointegration	Causality	
			Linear	Nonlinear
$GDP_{t}$ and $Total_{t}$	$CO2_{t}$	√ ***	√ ***	√ **
$CO2_{t}$ and $Total_{t}$	$GDP_{t}$	√ ***	×	√ **
$CO2_{t}$ and $GDP_{t}$	$Total_{t}$	√ ***	√ ***	√ **

√ indicates that there is relationship, while × indicates that there is no relationship. *, ** and *** denote significance at 10%, 5%, and 1% levels, respectively.

**Table 12 ijerph-16-04176-t012:** Summary of bivariate cointegration and causality results.

Independent Variable	Dependent Variable	Cointegration	Causality	
			Linear	Nonlinear
$GDP_{t}$	$CO2_{t}$	√ ***	√ ***	×
$Total_{t}$	$CO2_{t}$	√ ***	√ ***	√ *
$CO2_{t}$	$GDP_{t}$	√ ***	√ *	√ ***
$Total_{t}$	$GDP_{t}$	√ **	√ *	√ *
$CO2_{t}$	$Total_{t}$	√ ***	√ ***	×
$GDP_{t}$	$Total_{t}$	√ **	√ ***	×

√ indicates that there is relationship, while × indicates that there is no relationship. *, ** and *** denote significance at 10%, 5%, and 1% levels, respectively.

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
