# Peer review of "Modelling Economic Growth, Carbon Emissions, and Fossil Fuel Consumption in China: Cointegration and Multivariate Causality"

_ijerph, 2019, doi:10.3390/ijerph16214176_

Round 1

Reviewer 1 Report

This study aims to use the mathematical model to investigate the relationships among fossil fuel consumption, ??2 emissions, and economic growth, it is interesting and well-organized, and it can be accepted after revisions:

A comprehensive literature review is prerequisite to identify the research gaps. Sensitivity analysis or uncertainty analysis in the model? More policy implications.

Author Response

Dear Reviewer:

Thank you very much for your invaluable comments and feedback, which have helped to improve this manuscript significantly:

This study aims to use the mathematical model to investigate the relationships among fossil fuel consumption, emissions, and economic growth,

it is interesting and well-organized, and it can be accepted after revisions

Taking into consideration your helpful comments and suggestions, we have revised and improved the paper significantly. 

Comment 2:

A comprehensive literature review is prerequisite to identify the research gaps.

Reply: We have revised the "Literature Review" in the revised manuscript to identify the research gaps.

Comment 3: Sensitivity analysis or uncertainty analysis in the model?

Reply: We have discussed the issue in the revised manuscript.

Comment 4:

More policy implications.

Reply: We have included more policy implications in the revised manuscript.

Reviewer 2 Report

The topic of the paper has been largely debated in literature, and a great amount of studies has been devoted to this issue. However, it is still relevant, especially in regard to developing economies. The originality aspects of the paper are not clearly discussed. The sample selection, novelty of results and relevance of policy implications should be more clearly debated.

The literature review is partial and incomplete. You can refer to "The determinants of CO2 emissions in MENA countries: a responsiveness scores approach" Magazzino, C., Cerulli, G. 2019

A link to the database is absent.

Generally speaking, the econometric methodology is appropriate. However, the values of median, variance, IQR, CV, and range are missed. What about outliers and aberrant observations?

The results of additional unit root and stationarity tests should presented and discussed: 1: Leybourne; 2: Kapetanios-Shin; 3. Kapetanios-Shin-Snell; 4. DF-GLS; 5. KPSS; 6. ERS.

Policy implications are weak.

The editing needs a revision.

The English-style should be improved, since some grammatical errors are present, and some typos fixed.

Author Response

Dear Reviewer:

Thank you very much for your invaluable comments and suggestions, which have helped to improve this manuscript significantly:

The topic of the paper has been largely debated in literature, and a great amount of studies has been devoted to this issue.

it is still relevant, especially in regard to developing economies.

Generally speaking, the econometric methodology is appropriate.

According to your helpful comments and suggestions, we have revised and improved the paper significantly.

Comment 2:

The originality aspects of the paper are not clearly discussed.

Reply: We have discussed clearly the originality of the paper in the revised manuscript.

Comment 3:

The sample selection, novelty of results and relevance of policy implications should be more clearly debated.

Reply: We have included greater discussion on sample selection, novelty of the results, and relevance of policy implications in the revised manuscript.

Comment 4:

The literature review is partial and incomplete.

Reply: We have improved the “Literature Review” in the revised manuscript.

Comment 5:

A link to the database is absent.

Reply: We have added a link to the database.

Comment 6:

However, the values of median, variance, IQR, CV, and range are missed. What about outliers and aberrant observations?

Reply: We have displayed the values of the median, variance, IQR, CV, and range, which have been discussed in the revised manuscript. In addition, after using the univariate approach and outliers test, we find that there exist no outliers and aberrant observations in the sample data. We have included the discussion in the revised manuscript.

Comment 7:

The results of additional unit root and stationarity tests should be presented and discussed: 1: Leybourne; 2: Kapetanios-Shin; 3. Kapetanios-Shin-Snell; 4. DF-GLS; 5. KPSS; 6. ERS.

Reply: The Leybourne, Kapetanios-Shin, Kapetanios-Shin-Snell, DF-GLS, KPSS, and ERS tests have been presented and discussed in the revised manuscript.

Comment 8:

Policy implications are weak.

Reply: We have improved the discussion of policy implications in the revised manuscript.

Comment 9:

The editing needs a revision.

Reply: We have edited the paper carefully.

Comment 10:

The English-style should be improved, since some grammatical errors are present, and some typos fixed.

Reply: We have improved the English-style and corrected the grammatical errors and typos in the revised manuscript.

Reviewer 3 Report

Dear Authors,

the analysis of Modeling Economic Growth, Carbon Emissions, and Fossil Fuel Consumption in China: Cointegration and Multivariate Causality is very interesting and actual.

The following comments of the manuscript should be considered:

For the statements in lines 57 – 64 citation should be added – no source for the statements is available, Line 186 – a space is needed between the CO2 and emissions, Page 7 – the data sources for the analysis seems to be presented only on this page, but not in the References section – it would be useful to include these data sources into the References section too, Page 7 – the figure does not have any title. The title is only on page 8. This figure could be split and instead of 1 two figures could be produces. On the page 8 the Figure 1: some more detail should be given for this picture: Is it the GDP per capita? In USD? In USD PPP? In some real terms? – it is not clear from the figure 1. Line 316 – incorrect spelling: achievementsin China Line 317 – check the following: movietogether Line 347 – check the wording: is and again is (is – is) Line 446 – a space is needed between the CO2 and emissions, Page 14 – the equations are not numbered as before, Line 546 - a space is needed between the CO2 and emissions, Page 14 – instead of “Basis statistics” use the Basic statistics wording (for example in 5.1 title), Page 17 – check the equation (5.4): the value 1.2299 is in table 4 positive, but in equation negative, Lines 613 – 615: check the explanation: “with one percent increase in fosil…” should be changed to “CO2”. Page 15, line 574 and page 17, line 623: the same number 5.3 for different subheadings is given. Page 20 – in tables 10 and 11 – the word: “indicaties” is a mistake (the right word should be: indicates), Page 21 – are the presented hypotheses only the alternative hypotheses? Page 24 – line: 870 - check the symbols used for the hypotheses in this line with the symbols used in sections 6.1 and 6.2 line 916 – correct the following: “controlling??2emissions”. Page 26: is the following statement correct?: "Nonetheless, the causality analysis concludes that reducing the rates of both fossil fuel consumption and ??2 emissions simultaneously will cause economic growth" ?

In the paper some statements are difficult to follow. For example, on page 26: The Discussion 2. The wording could be sometimes easier. Using too many “negative expressions” in a sentence/sentences makes it difficult to track the analysis, the results.

The results and discussion of the study are interesting, useful for research.

Author Response

Dear Reviewer:

Thank you very much for your invaluable comments and suggestions, which have helped us to improve this manuscript significantly:

The analysis of Modeling Economic Growth, Carbon Emissions, and Fossil Fuel Consumption in China: Cointegration and Multivariate Causality is very interesting and actual.

The results and discussion of the study are interesting, useful for research.

We have revised and improved the paper significantly according to your insightful comments. 

Comment 2:

For the statements in lines 57 – 64 citation should be added – no source for the statements is available.

Reply: The citation has been added in the revised manuscript.

Comment 3:

Page 7 – the data sources for the analysis seems to be presented only on this page, but not in the References section – it would be useful to include these data sources into the References section too.

Reply: The data sources have been added into the References section in the revised manuscript.

Comment 4:

Page 7 – the figure does not have any title. The title is only on page 8. This figure could be split and instead of 1 two figures could be produces. On the page 8 the Figure 1: some more detail should be given for this picture: Is it the GDP per capita? In USD? In USD PPP? In some real terms? – it is not clear from the figure 1.

Reply: The title and more details have been given for the figure in the revised manuscript.

Comment 5:

Line 186 – a space is needed between the  and emissions, Line 446 – a space is needed between the  and emissions, Line 546 - a space is needed between the  and emissions, Line 316 – incorrect spelling: achievementsin China Line 317 – check the following: movietogether Line 347 – check the wording: is and again is (is – is) Page 14 – the equations are not numbered as before Page 14 – instead of “Basis statistics” use the Basic statistics wording (for example in 5.1 title), Page 17 – check the equation (5.4): the value 1.2299 is in table 4 positive, but in equation negative, Lines 613 – 615: check the explanation: “with one percent increase in fosil…” should be changed to “ ”. Page 15, line 574 and page 17, line 623: the same number 5.3 for different subheadings is given. Page 20 – in tables 10 and 11 – the word: “indicaties” is a mistake (the right word should be: indicates), Page 21 – are the presented hypotheses only the alternative hypotheses? Page 24 – line: 870 - check the symbols used for the hypotheses in this line with the symbols used in sections 6.1 and 6.2 line 916 – correct the following: “controlling emissions”. Page 26: is the following statement correct?: "Nonetheless, the causality analysis concludes that reducing the rates of both fossil fuel consumption and  emissions simultaneously will cause economic growth" ?

Reply: We have addressed the numerus helpful issues, and formatted and checked the revised paper carefully.

Comment 6:

In the paper some statements are difficult to follow. For example, on page 26: The Discussion 2. The wording could be sometimes easier. Using too many “negative expressions” in a sentence/sentences makes it difficult to track the analysis, the results.

Reply: We have revised, formatted and checked the paper carefully.